# Not only a helper, but also a teacher: Inter-active LLM Cascade

## Abstract

Large Language Models (LLMs) vary widely in their capabilities, with larger models often having better performance but higher cost: choosing an LLM model often involves trading off performance and cost. The *LLM Cascade* is a paradigm that *defers* difficult queries from weak/cheap to strong/expensive models. This approach is nonadaptive: the deferral decision rule is trained or derived by algorithms offline. When confronted with similar or repeated queries, the LLM Cascade may then repeatedly consult the expensive model and incur higher cost. To improve the cascading efficiency, we propose *Inter-Cascade*, an online and interactive LLM Cascade that extends the role of strong model from a backup helper to a long-term teacher. In our system, when a strong model resolves a difficult query, it also distills its solution into a generalized, reusable problem-solving strategy that boosts the weak model on subsequent queries. Adding strategies to queries enables the weak model to dynamically improve its performance over time, avoiding computationally and time-intensive fine-tuning. Empirically, compared with standard LLM Cascade baselines across multiple benchmarks, the Inter-Cascade significantly improves the accuracy of the weak model (by up to 33.06 absolute percentage points) and the overall system (by up to 5.53 absolute percentage points), while reducing the calls to strong models (by up to 48.05% relative reduction) and saving the corresponding fees (by up to 49.63% relative reduction). Inter-Cascade demonstrates the effective in-context knowledge transfer between LLMs, and provides a general, scalable framework applicable to both open-source and API-based LLMs.

## 1 Introduction

Large Language Models (LLMs) demonstrate remarkable performance across a wide range of generation and reasoning tasks. LLMs with stronger performance are generally larger in size, and the converse holds as well (Kaplan et al., 2020). Larger models often achieve better performance on more challenging tasks but are correspondingly more expensive. Depending on their expected workload, cost-sensitive users may wish to use *weaker* (and cheaper) models that suffice for simple queries and reserve the use of *stronger* (and expensive) models for more complex queries. In a prototypical example, a weaker model may run on a mobile device such as a phone while a stronger model may run in a cloud-based server owned by another service: the cost of the stronger model can include latency, monetary charges, or both. Since accuracy alone is not the only performance metric of interest, practical deployment scenarios require balancing multiple objectives including efficiency, latency, reliability, and network resource usage (Zhou et al., 2024; Khatun & Brown, 2024; Gundla & Atthuluri, 2025; Yan & Ding, 2025; Zhou et al., 2024).

The *LLM Cascade* has emerged as a widespread LLM paradigm in which weaker models handle routine queries and *defer* uncertain cases to stronger models in a sequential order (Chen et al., 2024). Deferral depends on a *deferral function*, typically estimates a confidence score that decides when to send queries to the strong model. The current approach focuses on improving this deferral decision by training or adjusting the threshold for the confidence score to decide when to defer (Shen et al., 2024; Rayan & Tewari, 2025; Ong et al., 2025; Zellinger et al., 2025; Zellinger & Thomson, 2025; Xia et al., 2024; Nie et al., 2024; Jung et al., 2025). The strong model, weak model, and deferral function are optimized prior to deployment. After training, the system follows the same

pipeline for any incoming query, which means processing is not adaptive to the workflow during inference/deployment. A static LLM Cascade pipeline would result in a substantial waste of tokens.

NVIDIA's recent position paper (Belcak et al., 2025) shows that many LLM applications repeatedly perform a small set of specialized tasks with only modest variations. There are various of scenarios that contain inherently similarities. For instance, datasets like GSM-Plus (Li et al., 2024), an extension of a math problem dataset GSM8K (Cobbe et al., 2021b), contains eight variants for each problem. Identical questions are repeatedly asked: e.g., "Which is larger, 9.9 or 9.11?" Users were once eager to keep asking this question to any newly released LLM (Korzhov, 2023; Schnabel, 2024; Junco, 2025). However, current LLM Cascade methods do not take this "similarity phenomenon" into consideration. As a result, a large amount of tokens are wasted because of repeated or similar queries. If a weak models consistently fails on similar or recurring problems, the system must repeatedly consult the strong model each time, which is wasteful/costly. The non-interactive nature of traditional LLM cascades, where weak models can only offload, prevents them from leveraging feedback from stronger models' capabilities during generation/inference. Furthermore, fine-tuning weaker models to overcome such failures is expensive and sometimes impractical: fine-tuning requires substantial memory (e.g., finetune Qwen3-235B requires 130 GB VRAM for LoRA or 2560 GB for full-parameter fine-tuning (Yang et al., 2025)) and must re-train again when distributions shift. For API-based models, fine-tuning may not even be available.

To take advantage of this phenomenon, in this paper we improve current "static" LLM Cascade by developing a new online adaptation method to assist the weaker model in dynamically balancing cost and accuracy. Our key insight is to exploit input similarity so the strong model can help the weaker one adapt and handle similar queries locally. The essence of the approach is to let the weaker model learn from the stronger model online: the weaker model can build a "crib sheet" using prior queries to do prompt engineering at the input that will guide the weaker model to the correct solution locally.

Our approach is influenced by in-context learning (ICL) (Dong et al., 2024) or few-shot prompting (Parnami & Lee, 2022), which can partially alleviate this limitation. In those approaches, by carefully selecting demonstrations or instructions one can enhance the reasoning capacity of weaker models without retraining. However, existing ICL and few-shot methods often rely on manual prompt design or retrieval from either fixed or manually updated database, making them inflexible as query distributions evolve. Our approach is related to Retrieval-Augmented Generation(RAG) (Lewis et al., 2020), in which a database is used for assisting the generation of answer. However, the databases in RAG studies are either built with engagement of human (Edge et al., 2025; Chen et al., 2025a; Shi et al., 2024) or updated from single LLM dialogue history for personalization usage (Zhang et al., 2025; Mo et al., 2025), while in our approach, the knowledge corpus are extracted from an extra stronger LLM without any human intervention. The goal in this work is to develop a framework the enables cascaded LLMs to interact adaptively: the weak model can benefit from *in situ* reasoning generated by the strong model to improve its own performance during real-world query streams. As a metaphor, the weak model uses the strong model for "on the job" training to improve longer-term performance. We further discuss this paradigm's relationship to extensive related works with details in the Appendix B.

**Primary contributions.** We improve on existing LLM Cascade approaches. **(1)** We propose a new framework, *Inter-Cascade*, for online and interactive LLM Cascades in which the strong model serves as both a backup helper and longer-term teacher. The strong model can both answer difficult queries and provide feedback that can be reused by the weak model to generalize problem-solving for future queries. These strategies are stored in a local database which the weak model uses as an auxiliary input: its performance is improved on similar future queries by leveraging the strategies generated from the strong model. In this way the strong LLM "teaches" the the weak LLM how to resolve these queries on its own. We think of this approach as a kind of *in-context knowledge distillation* approach to LLM Cascade using similarity-based memory. **(2)** We propose a theoretical model and show that without changing the deferral rule, adding strategies helps the weak LLM's confidence score better approximate its probability of correctness. This shows that using strategies can provably guarantee the higher accuracy of the overall system. **(3)** We compare to the LLM Cascade (Jung et al., 2025) and show that Inter-Cascade improves the overall system accuracy by up to 33.06 absolute percentage points and the overall system by up to 5.53 absolute percentage points, while reducing the usage of strong models by up to 48.05% relative reduction, This can reduce the corresponding fees by up to 49.63% relative reduction with the same guarantee of risk tolerance

and error level over all benchmarks. Crucially, our framework is general and modular: it applies to both API-only models and open-source models, and can be combined with any deferral function or any number of LLMs in cascade. We make the full implementation of Inter-Cascade under an open source license.

## 2 IMPROVING THE LLM CASCADE

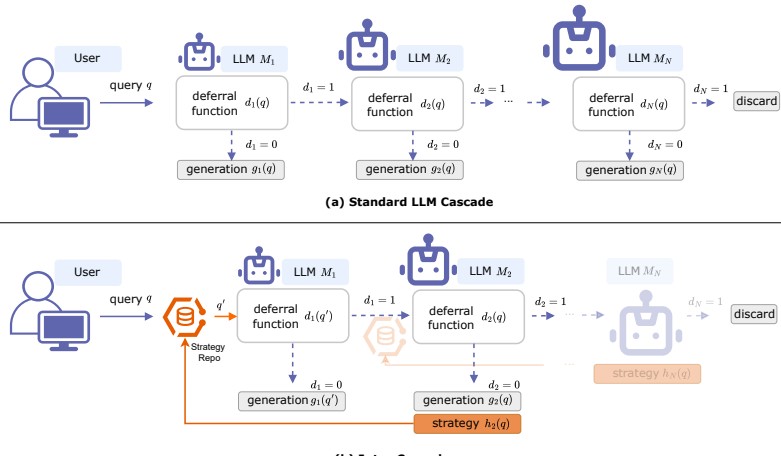

Figure 1: (a) Pipeline of standard LLM Cascade systems. (b) Pipeline of Inter-Cascade. The unique components in Inter-Cascade are painted in orange. For the sake of clarity and readability, we only present the case of two LLMs Inter-Cascade system and the scalable parts beyond two LLMs are rendered in a lighter color.

We first describe the standard LLM Cascade (Chen et al., 2024) and revisit the accuracy bound and calibration method for the deferral threshold proposed by Jung et al. (2025). We then introduce our proposed method Inter-Cascade and provide a theoretical framework to show when a weak model will be improved by a strong model's strategies.

### 2.1 STANDARD LLM CASCADE

Figure 1(a) shows the general $N$-LLM Cascade system (Chen et al., 2024). Each LLM $M_i : i \in [N]$ contains two key components. One is the *generation function* $g_i : \mathcal{Q} \to \mathcal{A}$, where $\mathcal{Q}$ is the space of queries and $\mathcal{A}$ is the space of answers. The other is *deferral function* $d_i : \mathcal{Q} \to \{0, 1\}$, which determines whether the $i$-th LLM will answer the query by itself ($d_i(q) = 0$) or defer it to the $(i+1)$-th LLM ($d_i(q) = 1$). Processing by the LLMs proceeds sequentially from $M_1$ to $M_N$. We define a partial order $\preccurlyeq_{\mathrm{wbc}}$ ("weaker but cheaper") to compare models (see Appendix C) and assume that in the cascade, $M_1 \preccurlyeq_{\mathrm{wbc}} M_2 \preccurlyeq_{\mathrm{wbc}} ... \preccurlyeq_{\mathrm{wbc}} M_N$. For each query $q \in \mathcal{Q}$, the first LLM $M_1$ takes the query $q$ and gives a final answer $g_1(q)$ if deferral function $d_1(q) = 0$, otherwise $M_1$ defers this query to the next LLM $M_2$ if $d_1(q) = 1$. If $M_2$ takes the query from $M_1$, it repeats the same process and so do the other LLMs except the last model $M_N$. As $M_N$ doesn't have another LLM to offload the query, $M_N$ discards this query if $d_N(q) = 1$. Recent studies propose different deferral functions $d_i$ to meet the demands in different scenarios. We focus on the two-LLM case in the rest of this paper, as shown in Figure 1(b). We call $M_1$ the *Weak LLM* and $M_2$ the *Strong LLM*. One common choice of deferral function is:

$$d_i(q) = \left\{ \begin{array}{ll} 0, & \text{if } c(q) \geq \lambda, \\ 1, & \text{otherwise,} \end{array} \right. \tag{1}$$

where $c : \mathcal{Q} \to [0, 1]$ is a pre-defined or pre-trained "confidence" metric (usually defined in terms of the probability of output tokens) and $\lambda$ is a confidence threshold, which is a hyperparameter that controls the trade-off between the system performance and cost.

**Accuracy Guaranteed LLM Cascade.** It is well known that LLMs suffer from systematic bias (Wang et al., 2024b; Thakur et al., 2025) and over-confidence (Xiong et al., 2024). To address this,

Jung et al. (2025) propose a post-hoc calibration algorithm, which provably guarantees that with the derived $\lambda$,

$$P\left(g_i(q) = a_{\text{true}} \mid c(q) \geq \lambda\right) \geq 1 - \alpha \tag{2}$$

with probability at least $1 - \delta$, as proved in Theorem 1 of their work, where $a_{\text{true}}$ is the ground-truth answer to query $q$. The risk tolerance $\alpha$ and error level $\delta$ are hyperparameters corresponding to the applications and users' demands. To instantiate this guarantee, a fixed-sequence testing (Bauer, 1991) procedure is first conducted, which selects the largest threshold $\lambda$ from a calibration set, such that $P\left(g_i(q) = a_{\text{true}} \mid c(q) \geq \lambda\right)$ is exactly and tightly bounded. The procedure is summarized in Algorithm 1. They also extend the single-model guarantees to the full cascade; see Section 2 and Appendix A.2 in Jung et al. (2025)'s paper for details.

---

**Algorithm 1** Calibrating Deferral Threshold $\lambda$ (Jung et al., 2025)

---

**Input:** Calibration set $(q, a) \in D_{\text{cal}}$, confidence metric $c(\cdot)$, risk tolerance $\alpha$, error level $\delta$
**Output:** Threshold $\lambda$
1: Initialize $\Lambda = \{0.999, 0.998, \ldots\}$ in decreasing order
2: **for** $\lambda \in \Lambda$ **do**
3:      $n(\lambda) \leftarrow \sum_{(q,a)\in D_{\text{cal}}} \mathbf{1}\{c(q) \geq \lambda\}$
4:      $\hat{R}(\lambda) \leftarrow \frac{1}{n(\lambda)}\sum_{(q,a)\in D_{\text{cal}}}\mathbf{1}\{g_i(q) \neq a_{\text{true}} \wedge c(q) \geq \lambda\}$
5:      $\hat{R}^+(\lambda) \leftarrow \sup\{R : \Pr[\text{Bin}(n(\lambda), R) \leq n(\lambda)\hat{R}(\lambda)] \geq \delta\}$
6:      **if** $\hat{R}^+(\lambda) \leq \alpha$ **then return** $\lambda$

---

The general pipeline of LLM Cascade is shown in Figure 1(a). By using this LLM cascade diagram, the deferral function can keep "confident" queries on Weak LLMs and only send "uncertain" queries to Strong LLMs, dramatically reducing at most 82.5% usage of the strongest LLM as shown by Jung et al. (2025) while ensuring the error rate is bounded by $\alpha$ with probability at least $1 - \delta$.

## 2.2 INTERACTIVE LLM CASCADE

LLM Cascade methods can be efficient and reliable although they still incur some waste in terms of tokens and latency as noted in Section 1. In particular, for workloads in which the Weak LLM is fed a similar or repeated queries for which it chooses to defer, the Strong LLM is called repeatedly to generate the same tokens. To address this issue, we propose *Inter-Cascade*. In Inter-Cascade, for both Weak LLM and Strong LLM, besides deferral function and generation function, we add the following components: *strategy generator* and *strategy repository*. In Strong LLM, we set up a *strategy generator* $h\colon \mathcal{Q} \to \mathcal{S}$, where $\mathcal{S}$ is the space of strategies. The strategy $s \in \mathcal{S}$ is defined as a sequence of tokens that contains the query and the answer of Strong LLM, together with a generalized ideas or tips to solve logically similar problems. To store those strategies, we construct a Strategy Repository called Repo. The Repo is accompanied by a *strategy matching function* $f\colon \mathcal{Q} \times \mathcal{Q}^N \to \mathcal{S}^k$, where $N$ is the size of current Repo and $k$ is a predefined hyperparameter that determines the number of strategies retrieved. The detailed description of strategy repository is depicted below:

**Strategy Repository.** The Strategy Repository Repo is formally defined as a collection of query-strategy pairs: $\text{Repo} = (q_j, s_j)_{j=1}^N$ where $q_j \in \mathcal{Q}$ are previously solved queries and $s_j \in \mathcal{S}$ are their corresponding strategies generated by Strong LLM. The strategy matching $f$ operates through multiple stages. The repository is initialized as an empty set and dynamically updated: when the Strong LLM generates a strategy $s = h(q)$ for a new query $q$, the pair $(q, s)$ is added to Repo, enabling future reuse through the matching function $f$.

For a query $q \in \mathcal{Q}$ that is sent to the Weak LLM, let $\text{sim}\colon \mathcal{Q} \times \mathcal{Q} \to [0, 1]$ be a ranking function. Let the Top-$k$ indices (sorted by decreasing similarity) be

$$\text{TopIndex}(q) \triangleq (t_1, t_2, \ldots, t_k),$$

where each $t_i \in \{1, \ldots, N\}$ indexes an item in Repo and $\text{sim}(q, q_{t_1}) \geq \cdots \geq \text{sim}(q, q_{t_k}) \geq \text{sim}(q, q_{\text{else}})$. After ranking, these strategies with Top-$k$ indexes are chosen to help the Weak LLM. Then the output of strategy matching function is $f(q, \text{Repo}) \triangleq \{s^{t_i} \mid t_i \in \text{TopIndex}(q)\}$.

*Remark* 2.1. Compared with finetuning or paying for Strong LLM, the cost of maintaining a $\mathrm{Repo}$ and running similarity-based matching algorithms are negligible. According to the estimate formula suggested by Johnson et al. (2021), conducting retrieval and Top-$2$ ranking on $1$ million query embeddings, which are $384$ dimensional vectors (the same size we used in experiments), only requires $0.2-0.8$ ms with $70-80$ MB GPU VRAM and $80-100$ MB RAM for long term storage. The demand can be easily fulfill on any PC or even phone, and imperceptible to human users.

**Inter-Cascade Pipeline.** The overall pipeline of Inter-Cascade is presented in Algorithm 2 and in Figure 1(b). For each query $q$, the Weak LLM first uses the strategy matching function $f(q, \mathrm{Repo})$ to find the most related strategies. The query and these strategies are then sent to deferral function. The augmented input is the prompt concatenation of query and strategies: $q' = [q, s^{t_1}, s^{t_2}, ..., s^{t_k}]$. If the Weak LLM's deferral function $d_1(q') = 0$, then final answer $a$ for current query is $g_1(q')$. If $d_1(q') = 1$, the query $q'$ is deferred to Strong LLM. Each time the query is sent to the Strong LLM, the deferral function in Strong LLM is called. If $d_2(q) = 0$, this query is discarded (since Strong LLM is the last model in two LLMs Cascade), otherwise $g_2(q)$ produces the answer and further, a new strategy is produced by $h(q)$. Then, the strategy will be stored into $\mathrm{Repo}$. Given $\alpha$ and $\delta$, we can derive the $\lambda$ from Algorithm 1 and determine deferral function $d_1$ and $d_2$ as defined by (1). Our algorithm can be extended to multi-LLM cases, the corresponding Algorithm 3 is shown in Appendix D.

---

**Algorithm 2** Inter-Cascade Inference Pipeline

---

**Input:** Test set $\mathcal{T} = \{q_1, \ldots, q_I\} \subseteq \mathcal{Q}$; Weak LLM with deferral function $d_1$, generation function $g_1$, strategy repository $\mathrm{Repo} = \emptyset$; strategy matching function $f$; Strong LLM with deferral $d_2$, generator $g_2$, and strategy generator $h$.
  **Deferral convention:** $0 =$ handle locally, $1 =$ defer/forward.
1: **for** $i \leftarrow 1$ **to** $I$ **do**
2: $\quad [s_i^{t_1}, s_i^{t_2}, ..., s_i^{t_k}] \leftarrow f(q_i, \mathrm{Repo})$ $\qquad\qquad$ ▷ **Top-$k$ strategies matching from** $\mathrm{Repo}$
3: $\quad q_i' \leftarrow [q_i, s_i^{t_1}, s_i^{t_2}, ..., s_i^{t_k}]$ $\qquad\qquad\qquad$ ▷ concatenate query and strategies
4: $\quad$ **if** $d_1(q_i') = 0$ **then** $\qquad\qquad\qquad\qquad\qquad\qquad$ ▷ **Weak LLM decision**
5: $\quad\quad$ generate answer $a_i \leftarrow g_1(q_i')$ $\qquad\qquad\qquad$ ▷ Answer locally at Weak LLM
6: $\quad$ **else**
7: $\quad\quad$ **if** $d_2(q_i) = 0$ **then** $\qquad\qquad\qquad\qquad\qquad\qquad$ ▷ **Defer to Strong LLM**
8: $\quad\quad\quad$ $s_{\mathrm{new}} \leftarrow h(q_i)$ $\qquad\qquad\qquad$ ▷ Strong LLM synthesizes a new strategy
9: $\quad\quad\quad$ $\mathrm{Repo} \leftarrow \mathrm{Repo} \cup \{(q_i, s_{\mathrm{new}})\}$ $\quad$ ▷ Send back strategy to Weak LLM and store
10: $\quad\quad\quad$ generate answer $a_i \leftarrow g_2(x_i)$ $\qquad\qquad\qquad$ ▷ Answer at Strong LLM
11: $\quad\quad$ **else**
12: $\quad\quad\quad$ Discard current query $q_i$ $\qquad$ ▷ None of LLMs are confident to answer the query

---

**Strategies Provide Improved Calibration.** The $\mathrm{Repo}$ we build during the usage of the combination of LLMs collects the strategies of the Strong LLM and provides strategies to help the Weak LLM answer queries. With the help of strategies, the Weak LLM is able to solve the more challenging problems that appear frequently and be more aware of its correctness of answering the queries, leading better confidence. However, it is not clear that how this increment in the accuracy and the quality of confidence could be preserved in the queries after the filtration. After all, all the queries, even to which the Weak LLM answers correctly would be deferred if the Weak LLM's confidence can not pass the threshold. Therefore, we present the following theories to estimate such an increment that would remain in the filtered queries.

To be specific, we first assume that, after adding strategies, under the same confidence threshold $\lambda$, the number of queries that pass the confidence threshold increases from $n(\lambda)$ to $bn := n'(\lambda)$, $b \in [1, \infty)$, where $n(\lambda)$ is first defined in Algorithm 1. The number of wrongly answered queries before and after the help of strategies are denoted by $x$ and $\epsilon x$, respectively, where $\epsilon \in (0, 1)$. We want to understand the potential benefit in terms of the reduction in risk $\alpha$ under the same error level $\delta$. We do not change the threshold $\lambda$, which is the case when the strategy repository is enlarged during the running process of the Inter-Cascade. Theorem 2.2 states our main result. For the convenience of the statement, we define $\alpha(\epsilon, b)$ as the value of risk tolerance $\alpha$ when total number of queries that pass threshold is $bn$ and incorrectly answered queries is $\epsilon x$.

**Theorem 2.2.** *Suppose that $\widehat{R}^+(\lambda)$ is a monotonic decreasing function of $\lambda$. Fix $\delta \in (0,1)$ and an integer $n \geq 1$. For $x \in \{0, 1, \ldots, n\}$, $\epsilon \in (0, 1]$, and $b \in [1, \infty)$. Suppose that $\min\{\epsilon x + 1, \, n - \epsilon x\}$ is moderately large and $1 - \delta$ is not an extreme tail, then:*

*(a) Decrease in value.* $\alpha(\epsilon, b) \leq \alpha(1, 1)$ *when $\epsilon \in (0, 1]$ and $b \in [1, \infty)$.*

*(b) Normal approximation for the amount of decrease.* *Let $z := \Phi^{-1}(1 - \delta)$, where $\Phi$ is the Normal cumulative distribution function, when $n$ is large enough, the decrease of the risk under same level of tolerance is given by,*

$$\alpha(1,1) - \alpha(\epsilon, b) \approx \left( \frac{x+1}{n+1} - \frac{\epsilon x + 1}{bn + 1} \right) + z \left[ \sqrt{\frac{(x+1)(n-x)}{(n+1)^2(n+2)}} - \sqrt{\frac{(\epsilon x + 1)(bn - \epsilon x)}{(bn+1)^2(bn+2)}} \right].$$

The proof of this theorem is in Appendix F. Theorem 2.2 states that, when the $\delta$ and confidence threshold $\lambda$ do not change, if more queries can pass the threshold, after combining with strategies and under certain conditions, we can ensure a smaller risk tolerance $\alpha$ in the guarantee of this inequality (2). That is, Inter-Cascade yields a higher success rate for Weak LLM.

Other than the case that $\lambda$ remains unchanged, which is analyzed above, another case may be that when the users want the same number of queries to be covered by the Weak LLM during two rounds of queries (before and after adding strategies). This case considers the influence of a better Weak LLM on our pipeline. In this case, we instead assume that $n(\lambda) = n(\lambda')$, which ensures the same coverage of Weak LLM. We also show that we can ensure a smaller risk tolerance $\alpha$ when threshold becomes $\lambda'$ while $\delta$ and number of queries that pass threshold remain unchanged. And the reduction in tolerance level $\alpha(1,1) - \alpha(\epsilon, 1)$ is approximately linear to $1 - \epsilon$. The full statement of Theorem G.1 and the proof are shown in Appendix G.

## 3 EXPERIMENTS

### 3.1 BENCHMARKS

In our experiments, we use two categories of datasets. The first category consists of reasoning-focused scientific benchmarks, including *GSM-Symbolic* (Mirzadeh et al., 2025), *GSM-Plus* (Li et al., 2024), and *MetaMath* (Yu et al., 2024), selected to evaluate performance on tasks requiring logical reasoning. The second category includes factual benchmark, represented by *NASA-History-MCQ* (Fleith, 2025), chosen to assess performance on tasks with lower reasoning demands. Using both categories allows for a more comprehensive assessment across tasks of different difficulty levels and types. The detailed descriptions of selected benchmarks are in Appendix I. The prompt template and an example problem for each benchmark are provided in Appendix L.

### 3.2 EXPERIMENTAL SETTINGS

**Inter-Cascade.** On all benchmarks, *Gemini-2.0-flash* consistently outperforms *GPT-3.5-turbo* (see ICTable 1), and is therefore designated as the Strong LLM in our two-LLM Inter-Cascade, with *GPT-3.5-turbo* as the Weak LLM. We extract the normalized token probability from the LLM's output as confidence score $c(q)$ in following experiments. In preparation phase, with given risk tolerance $\alpha$ and error level $\delta$, we derive desired confidence threshold $\lambda$ from calibration set by following Algo. 1. Then deploy corresponding deferral functions $d_i$ according to equation (1).

Our similarity-based strategy matching process on $\mathrm{Repo}$ works as follows. Given a new query, it is encoded into a vector and used to retrieve the top-$k$ semantically similar queries from $\mathrm{Repo}$. We employ the *all-MiniLM-L6-v2* transformer (Reimers & Gurevych, 2019) to produce $384$-dimensional sentence embeddings and use the FAISS library (Douze et al., 2025) for efficient approximate nearest-neighbor search. FAISS returns the top-$k$ vectors that minimize cosine distance, providing the Inter-Cascade with prior Strong LLM responses, including queries, answers and strategies, which can inform the Weak LLM's responses.

**Inter-Cascade with Random Strategies.** To evaluate the impact of similarity-based retrieval on $\mathrm{Repo}$, we randomly select the same number of strategies for each query, instead of choosing the top-$k$ most similar queries.

Table 1: Accuracies of the base LLMs on four benchmarks

| Dataset | LLM | Accuracy | Dataset | LLM | Accuracy |
|---------|-----|----------|---------|-----|----------|
| GSM-Symbolic | gpt-3.5-turbo | 13.36% | MetaMath | gpt-3.5-turbo | 37.30% |
| | gemini-2.0-flash | 69.36% | | gemini-2.0-flash | 79.70% |
| GSM-Plus | gpt-3.5-turbo | 23.00% | NASA-History | gpt-3.5-turbo | 65.30% |
| | gemini-2.0-flash | 73.57% | | gemini-2.0-flash | 78.80% |

**Jung Proposed LLM Cascade.** To evaluate the performance and effectiveness of the Inter-Cascade, we choose Jung et al. (2025)'s *Cascaded Selective Evaluation* as the baseline model. Its method for deriving confidence scores and thresholds provides a provable lower bound on the error risk and achieves state-of-the-art performance compared with other confidence-based LLM cascades.

## 3.3 EVALUATION METRICS

We first define the notations used in our evaluation. Let $T$ and $U$ denote the total number of queries and the number of uncovered queries in a benchmark, respectively. Let $N_w$ and $N_s$ be the number of times the Weak and Strong LLMs are invoked, and let $C_w$ and $C_s$ denote the number of queries correctly answered by these models that also pass the confidence threshold. $C_w^{\text{total}}$ denotes the total number of queries answered correctly by the Weak LLM. Let $\text{Tok}_J$ and $\text{Tok}_O$ be the tokens consumed by Jung's method and our proposed Inter-Cascade pipeline, and $\text{Cost}_J$ and $\text{Cost}_O$ denote their corresponding costs. The evaluation metrics are summarized in Table 2.

Table 2: Evaluation Metrics

| Metric | Formula | Metric | Formula |
|--------|---------|--------|---------|
| Pipeline Accuracy | $(C_w + C_s)/(T - U)$ | Strong LLM Call Rate | $N_s/T$ |
| Weak LLM Accuracy | $C_w^{\text{total}}/(T - U)$ | Weak Correct Accepted | $C_w/(T - U)$ |
| Coverage Rate | $(T - U)/T$ | Token Reduction | $(\text{Tok}_J - \text{Tok}_O)/\text{Tok}_J$ |
| Cost Reduction | $(\text{Cost}_J - \text{Cost}_O)/\text{Cost}_J$ | | |

## 3.4 PERFORMANCE AND COST ANALYSIS

**Inter-Cascade vs. Jung's LLM Cascade.** We evaluate our *Inter-Cascade* pipeline and Jung's method, as shown in Table 3. Our method outperforms Jung's, with a $4.33\% - 6.35\%$ increase in Pipeline Accuracy on reasoning benchmarks and a $0.76\%$ increase on the non-reasoning factual NASA-History benchmark. The Strong LLM Call Rate is reduced on all benchmarks, with reductions ranging from $4.41\%$ to $28.53\%$. These results indicate that *Inter-Cascade* pipeline is beneficial across different categories of tasks and particularly effective for reasoning-intensive tasks. Experiment results on extensive and diverse benchmarks are attached in Appendix J.

**Effectiveness of Similarity-Based Retrieval.** To isolate the effect of strategy selection, we include a control variant in which *Inter-Cascade* selects strategies at random. Across datasets, its performance generally falls between the *Inter-Cascade* and Jung's pipeline (see Table 3), demonstrating the benefit of similarity-based retrieval. Although outside the scope of this work, one possible future direction is to further refine the selection of strategies, which would involve verifying whether all of the top-$k$ retrieved strategies are relevant to the given queries. The accuracy of the *Inter-Cascade* (random strategies) differs from Jung's by $-2.43\%$ to $+1.93\%$, and its Strong LLM Call Rate shows only a modest reduction, ranging from $1.59\%$ to $5.17\%$.

**Impact of Inter-Cascade on Weak LLM.** Having examined the overall pipeline improvements, including Pipeline Accuracy and Strong LLM Call Rate reduction, we now investigate how our proposed *Inter-Cascade* affects the Weak LLM. As shown in Table 4, our Weak LLM outperforms the Weak LLM in the other pipeline across all benchmarks. The improvements are particularly pronounced on reasoning benchmarks, with gains of $23.21\%$, $16.2\%$, and $33.06\%$ on MetaMath, GSM-Plus, and GSM-Symbolic, respectively, while still achieving an improvement of $0.48\%$ on the non-reasoning NASA-History benchmark. Importantly, improvements in the Weak LLM's accuracy contribute to the pipeline's performance only when the correctly answered queries exceed the confidence threshold. This is captured by the *Weak Correct Accepted* metric in Table 4, which

Table 3: Results across datasets using different pipelines. "Jung" denotes Jung's LLM-Cascade and "Our (Retrieval)" denotes the Inter-Cascade with similarity-based retrieval. The number of strategies is fixed at $k = 2$ for both Inter-Cascade settings. Metrics reported are Pipeline Accuracy (Pipeline Acc.), Strong LLM Call Rate (Strong Call), and Coverage Rate (Cov.). (a) GSM-Symbolic: For the Strong LLM, $\alpha_s = 0.2, \delta_s = 0.8, \lambda_s = 0.47$. For the Weak LLM, $\alpha_w = 0.6, \delta_w = 0.6, \lambda_w = 0.45$. (b) GSM-Plus: For the Strong LLM, $\alpha_s = 0.2, \delta_s = 0.8, \lambda_s = 0.51$. For the Weak LLM, $\alpha_w = 0.6, \delta_w = 0.6, \lambda_w = 0.48$. (c) MetaMath: No threshold is applied for the Strong LLM. For the Weak LLM, $\alpha_w = 0.4, \delta_w = 0.6, \lambda_w = 0.61$. (d) NASA-History: No threshold is applied for the Strong LLM. For the Weak LLM, $\alpha_w = 0.2, \delta_w = 0.7, \lambda_w = 0.87$.

| Data | Pipeline | Pipeline Acc. (%)↑ | Strong Call (%)↓ | Cov. (%) |
|---|---|---|---|---|
| GSM-Symb. | Jung | 66.04 | 59.37 | 86.31 |
| | Our (Retrieval) | **70.37** | **30.84** | **90.35** |
| GSM-Plus | Jung | 52.78 | 46.29 | 93.57 |
| | Our (Retrieval) | **58.31** | **32.44** | **94.79** |
| MetaMath(20K) | Jung | 65.21 | 49.26 | 100.00 |
| | Our (Retrieval) | **71.56** | **23.68** | 100.00 |
| NASA-Hist. | Jung | 71.88 | 26.68 | 100.00 |
| | Our (Retrieval) | **72.64** | **22.54** | 100.00 |

Table 4: Results on Weak LLM across datasets. Reported metrics are Weak LLM Accuracy (Weak Acc.) and Weak Correct Accepted (Weak Corr. Accpt.). Parameter settings are the same as in Table 3.

| Data | Pipeline | Weak Acc. (%)↑ | Weak Corr. Accpt. (%)↑ |
|---|---|---|---|
| GSM-Symb. | Jung | 15.04 | 12.34 |
| | Our (Retrieval) | **48.10** | **46.09** |
| GSM-Plus | Jung | 22.46 | 19.13 |
| | Our (Retrieval) | **38.66** | **35.73** |
| MetaMath(20K) | Jung | 34.95 | 28.54 |
| | Our (Retrieval) | **58.16** | **54.07** |
| NASA-Hist. | Jung | 66.22 | 55.37 |
| | Our (Retrieval) | **66.70** | **58.40** |

represents the proportion of correctly answered queries that surpass the Weak LLM's threshold. The observed increase in Weak Correct Accepted shows that Inter-Cascade enhances not only the Weak LLM's accuracy but also its confidence on correct predictions, a crucial factor in converting local improvements into overall pipeline gains.

Table 5: Token and API cost changes across datasets for Inter-Cascade compared with Jung's pipeline.

| Benchmark | Weak LLM Tokens | | | Strong LLM Tokens | | | Token Price |
|---|---|---|---|---|---|---|---|
| | Total | Input | Output | Total | Input | Output | |
| GSM-Symb. | +147.66% | +148.80% | -17.10% | -47.80% | -45.80% | -51.32% | -49.63% |
| GSM-Plus | +145.96% | +147.11% | -3.56% | -29.95% | -29.51% | -30.90% | -30.41% |
| Meta.(20K) | +127.90% | +128.66% | -1.38% | -52.18% | -52.20% | -52.12% | -52.15% |
| NASA-Hist. | +132.58% | +133.40% | 0.99% | -15.47% | -15.22% | -16.07% | -15.75% |

**Effect of Strategies on Accuracy and Confidence Calibration.** As mentioned earlier, one notable observation from our experiments is that providing strategies enhances the Weak LLM's ability to assess its own accuracy. To further investigate this observation, we present Figure 2 for the GSM-Symbolic dataset. Analyses for the other three datasets, which exhibit similar patterns, are provided in Appendix H. Figure 2a depicts the accuracy of the Weak LLM as a function of the confidence threshold. For each threshold, only queries with confidence equal to or above the threshold are considered, and accuracy is calculated as the proportion of correct predictions. The figure further demonstrates that our pipeline consistently improves the accuracy of queries that pass the threshold. Figures 2b, 2c, and 2d illustrate the distribution of query confidence. The histogram offers insight into prediction coverage across different confidence thresholds and shows that our method outperforms the baselines in terms of coverage. Together, these figures indicate that our method not only helps the Weak LLM produce correct answers, but also enables it to better calibrate its confidence by being more confident when the answer is correct and less confident when it is incorrect.

Table 6: Processing Latency and Strategy Repository Size across different datasets. Retrieval refers to the time spent on strategies matching and ranking. Generation refers to time spent on generating answer via API.

| Benchmark | Tested Samples | Our | | | Jung | Repository Size |
|---|---|---|---|---|---|---|
| | | Total | Retrieval | Generation | Total | |
| GSM-Symb. | 11250 | 2.19s | 0.10s | 2.09s | 1.83s | 15.4 MB |
| GSM-Plus | 9504 | 1.72s | 0.06s | 1.66s | 1.66s | 12.9 MB |
| MetaMath(20K) | 20000 | 1.60s | 0.06s | 1.54s | 1.54s | 19.6 MB |
| NASA-Hist. | 6469 | 1.28s | 0.07s | 1.21s | 1.30s | 8.8 MB |

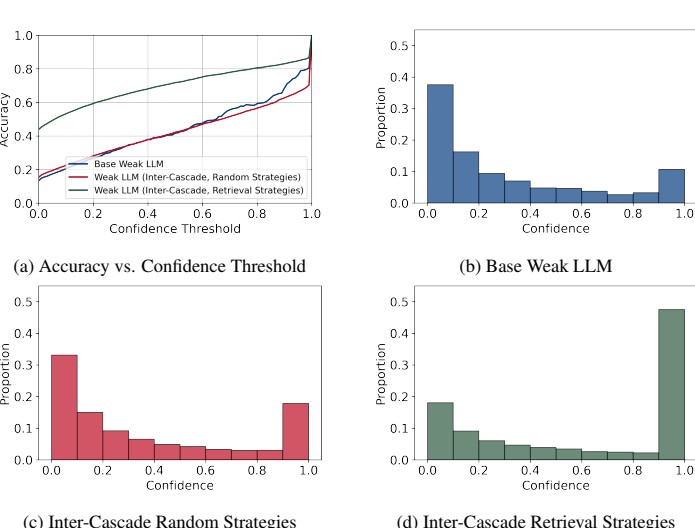

(a) Accuracy vs. Confidence Threshold

(b) Base Weak LLM

(c) Inter-Cascade Random Strategies

(d) Inter-Cascade Retrieval Strategies

Figure 2: GSM-Symbolic dataset: (a) Accuracy as a function of the confidence threshold for the base Weak LLM, Inter-Cascade with random strategies, and Inter-Cascade with retrieval strategies, and (b) - (d) their corresponding confidence histograms. Our Inter-Cascade (Retrieval) consistently concentrates probability mass near high confidence ($0.9 - 1.0$), while the weak and random variants place more mass at low confidence, which explains the accuracy gains observed in (a).

**Token and API Cost Savings.** Our pipeline not only improves accuracy but also reduces the number of Strong LLM calls, resulting in substantially lower token consumption on Strong LLM. Table 5 shows the percentage changes in token usage and corresponding API costs compared with Jung's pipeline. Table 6 shows the average processing time per query (including the call of Strong LLM) and the final size of strategies repository across datasets. The results imply that the time difference is between -0.02s and +0.36s, which won't impact the user experience. The size of repository is at level of 10MB+ when the number of queries is at 10K+ level, which can be easily maintained in resource limited settings like mobile or edge device. More promisingly, accumulated queries and responses can serve as training data for periodic offline fine-tuning the Weak LLM (for example as part of a software update), enabling a self-improving pipeline that dynamically adapts to new data.

**Ablation Study on Strategy Selection** In order to evaluate the impact of each part when we add strategies to the input of Weak LLM, we conduct ablation experiments for different settings: only adding similar questions and answers (No strategy), adding randomly selected strategies (Random), and our standard Inter-Cascade pipeline (Retrieval). The results in Table 7 and Table 8, show that the performance of Random Strategy method is between our standard pipeline and Jung's method, while No Strategy is not an acceptable option. Although in benchmarks like NASA-History, the overall accuracy is 2.00% higher than our standard pipeline, the cost is significant: the Strong Call Rate increase by 42.58%, which means only add similar question and answer to the input of Weak LLM would use 2.89x of the Strong LLM. Moreover, the Weak LLM's accuracy would be dramatically undermined by adding non-strategy information to the input of Weak LLM compared to the accuracy for single Weak LLM in Table 1. Only adding retrieved question and answers without instructive and generalized problem solving strategy to Weak LLM input is harmful: not only lower the accuracy of Weak LLM, but also call more Strong LLM, which is more expensive. Extensive Ablation studies on cold start of the strategy repository, effect of the size of strategies and different selection of LLM pairs are attached in Appendix K.

Table 7: Pipeline Accuracy and Strong LLM Call Rate in the ablation study on strategy selection : Our (No strategy) vs. Our (Random) vs. Our (Retrieval). Parameter settings are the same as Table 3.

| Data | Pipeline | Pipeline Acc. (%)↑ | Strong Call (%)↓ | Cov. (%) |
|------|----------|-------------------|------------------|----------|
| GSM-Symb. | Our (No strategy) | 67.55 | 65.15 | 83.14 |
| | Our (Random) | 63.61 | 54.20 | 87.90 |
| | Our (Retrieval) | **70.37** | **30.84** | **90.35** |
| GSM-Plus | Our (No strategy) | 58.12 | 54.81 | 93.83 |
| | Our (Random) | 53.63 | 43.64 | 94.10 |
| | Our (Retrieval) | **58.31** | **32.44** | **94.79** |
| MetaMath(20K) | Our (No strategy) | **74.48** | 57.32 | 100.00 |
| | Our (Random) | 67.85 | 45.99 | 100.00 |
| | Our (Retrieval) | 71.56 | **23.68** | 100.00 |
| NASA-Hist. | Our (No strategy) | **74.64** | 65.12 | 100.00 |
| | Our (Random) | 71.32 | 25.09 | 100.00 |
| | Our (Retrieval) | 72.64 | **22.54** | 100.00 |

Table 8: Weak LLM performance in the ablation study on strategy selection: Our (No strategy) vs. Our (Random) vs. Our (Retrieval). Parameter settings are the same as Table 3.

| Data | Pipeline | Weak Acc. (%)↑ | Weak Corr. Accpt. (%)↑ |
|------|----------|----------------|------------------------|
| GSM-Symb. | Our (No strategy) | 10.23 | 17.08 |
| | Our (Random) | 17.40 | 15.27 |
| | Our (Retrieval) | **48.10** | **46.09** |
| GSM-Plus | Our (No strategy) | 20.20 | 17.08 |
| | Our (Random) | 25.51 | 22.38 |
| | Our (Retrieval) | **38.66** | **35.73** |
| MetaMath(20K) | Our (No strategy) | 33.40 | 28.38 |
| | Our (Random) | 38.64 | 32.66 |
| | Our (Retrieval) | **58.16** | **54.07** |
| NASA-Hist. | Our (No strategy) | 28.21 | 22.88 |
| | Our (Random) | 65.22 | 55.56 |
| | Our (Retrieval) | **66.70** | **58.40** |

**Inter-Cascade Robustness under Automatic Strategies.** All strategies and their corresponding answers are generated by the Strong LLM in a streaming manner, and any strategy whose confidence exceeds the threshold $\lambda_s$ is automatically accepted. This differentiates *Inter-Cascade* from other LLM augmentation methods such as manually selected in-context learning, few-shot prompting, or static retrieval-augmented generation. Consequently, the strategy repository may contain incorrect strategies. Nonetheless, the results in Table 3 and Table 4 demonstrate the effectiveness of $\lambda_s$ and the robustness of the *Inter-Cascade* pipeline.

## 4 CONCLUSION

We propose *Inter-Cascade*, an online and interactive *LLM Cascade* framework that enables Weak LLM to learn online from Strong LLM's prior experience without fine-tuning. Inter-Cascade improves both the accuracy of Weak LLM and the overall system, while reducing the reliance on Strong LLM, saving computation, monetary cost, and latency (when Strong LLM is deployed on remote server), compared with current LLM Cascade.

Inter-Cascade provides a general and scalable framework for multi-LLM systems, which can be implemented with different LLMs and cascade layers. Despite the promising performance of Inter-Cascade, further improvements can still be achieved by proposing better methods of generating strategy, better algorithms to evaluate similarity and mechanisms to prevent mismatch in future work. Inter-Cascade is also naturally well-suited for distributed systems, where local Weak LLM owners can teach and boost their model in a tailored way by sending customized queries to Strong LLM. Another future work direction arises from Inter-Cascade's potential to bridge online and offline learning. While augmenting the system performance by incorporating queries with related strategies during online operation, the generated strategy repository can be exported as local training set for periodic finetuning, permanently improving the capability of Weak LLM. We hope Inter-Cascade inspires future research on building more interactive LLM Cascades or other multi-LLM systems.

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

## A    CLARIFICATION: USE OF LLMS ON AIDING OR POLISHING WRITING

We used ChatGPT and Gemini solely as writing assistants to correct the typos and grammars, help polish the language, improve clarity, and refine the presentation of this manuscript. The LLMs did not contribute to the conception of ideas, design of experiments, execution of analyses, or interpretation of results.

## B    EXTENDED RELATED WORK

**LLM Cascade** There are many LLM paradigms that contain collaboration between multiples LLMs in a system (Chen et al., 2025b): a)Ensemble before inference, where router choose one LLM from candidates for inference; b)Ensemble during inference, where LLMs work in parallel; c) Ensemble after inference, where LLMs work in sequence and LLM Cascade belongs to this filed.  LLM Cascade is firstly proposed by Chen et al. (2024) to balance the LLM performance and cost by allocating queries to a weak model or a strong model according to the confidence estimate of the queried question.  Shen et al. (2024) propose a latent variable model to let the weak model learn the deferral function at the token-level.  Rayan & Tewari (2025) also extend the Learning to Defer (Madras et al., 2018) setting to LLM by training a post-hoc deferral function for each token of the sequence.  Ong et al. (2025) train a separate router such that deferral decision can be made before sending the query to weak LLM, saving more tokens.  Zellinger et al. (2025) provide extra option to early discard the unsolvable queries in weak model.  Xia et al. (2024); Nie et al. (2024) formulate LLM Cascade as online problem to dynamically adjust its deferral policy over time. Zellinger & Thomson (2025) propose a rational tuning pipeline for LLM Cascade via probabilistic modeling.  Since the deferral result relies on the confidence score of weak model, there are are literatures focusing on boosting the the measure of confidence of weak model's output (Jitkrittum et al., 2023; Chuang et al., 2025).  Together with experimental verification,  Jung et al. (2025) conduct fixed sequence testing to provably guarantee the lower bound of accuracy.  Therefore, we choose Cascaded Selective Evaluation by Jung et al. (2025) as the baseline of our work.  Beside deferring to strong model, Beyond standard LLM Cascade,  Strong et al. (2025a) propose a deferral system that weak model also sends its generated intelligent guidance to strong model once deferred, boosting the performance of next level model.  However, current LLM Cascades cannot adapt to the query streaming once trained and deployed.  And the weak model cannot learn from the previous deferrals and corresponding strategies generated by the strong model, causing the waste of computation, tokens, money and sometimes communication.

**Learning With Reject Option** The general framework that allows a machine learning model to abstain from making decision was originally propose by Chow (1957; 1970) in the 1950s.  After decades, the Learning with reject option was continuously explored in different periods by Herbei & Wegkamp (2006) and Cortes et al. (2016).  The more recent works extend the framework to a multi models system where the local model can learn to defer its task to one expert (human or existing model)  (Madras et al., 2018; Mozannar & Sontag, 2020; Verma & Nalisnick, 2022; Mao et al., 2024b), multiple experts (Verma et al., 2023; Mao et al., 2024a) or unknown experts (Nguyen et al., 2025a; Strong et al., 2025b; Tailor et al., 2024).  There are literature that also explore the case when expert can learn to adaptively help the local model (Wu & Sarwate, 2024; Wu et al., 2025).  Adding reject option at the network layer level is another branch of works called early exiting (Teerapittayanon et al., 2016).  However, most of the learning with reject option works focus on classical prediction tasks, few of them address the NLP tasks that rely on generative-based model while this work focus on the collaboration between LLMs.

**Knowledge Distillation** Knowledge distillation (KD) is a machine learning technique for training smaller "student" models by transferring "knowledge" from larger, more powerful "teacher" models.  Classical knowledge distillation use soft-labels (Hinton et al., 2015) to let the student model learn the distribution of teacher model. The concept of KD is expanded to more levels: besides mimicking the output of teacher model, the student model can also learn from intermediate features (Romero et al., 2015; Pham et al., 2024), relationships (Joshi et al., 2024), actively chosen sample (Liu et al., 2024), principle discovery (Wang et al., 2024a) and itself  (Lee et al., 2023).  Our Inter-Cascade also helps the knowledge transfer from the Strong LLM to Weak LLM. However, current knowledge distillation relies on the training or finetuning of the student model and can not continue learning process during inference phase while our method doesn't require the updating of the LLM param-

eters and continually improves during the inference phase via dynamically matching stored Strong LLM's strategy.

**Retrieval-Augmented Generation(RAG)**RAG (Lewis et al., 2020) is an approach that combines pre-trained parametric and non-parametric memory for language generation. Given the focus of our work, we group RAG-style approaches into three categories: static RAG, history-aware RAG, and agentic RAG.

*Static RAG.* Classical RAG assumes a fixed, pre-constructed external corpus and focuses on how to retrieve, re-rank, and fuse evidence to support generation. Works in this line focus on design dense retrieval and re-ranking pipelines over a static collection (Lewis et al., 2020; Edge et al., 2025; Wang et al., 2025a; Rubin et al., 2022; Margatina et al., 2023).In all these methods, the source of knowledge is an offline, human-curated dataset, and the system's adaptivity lies purely in how it accesses this corpus, not what the corpus contains. By contrast, Inter-Cascade does not assume any pre-existing database: the "corpus" is constructed online as the strong LLM generates strategies and reasoning traces that are stored for future reuse by the weak LLM. Thus, our system is closer to an online, LLM-driven knowledge construction mechanism than to classical static RAG.

*History-Aware RAG.* A second line of work augments RAG with dialogue history and user feedback, dynamically updating a memory store based on past interactions. Conversational RAG frameworks like DH-RAG (Zhang et al., 2025), CHIQ (Mo et al., 2025) maintain short-term and long-term memories of successful dialogue turns, using them to improve future retrieval and personalization. Other methods such as ComRAG (Chen et al., 2025a), ERAGent (Shi et al., 2024), Pistis-RAG (Bai et al., 2024), and Social-RAG (Wang et al., 2025b) update user profiles or QA memories when users provide explicit positive feedback or when high-quality answers are validated by the social community. Despite their dynamism, these systems either take history information for self usage or treat the human user (or user community) as the source of new content. The resulting models are primarily personalized assistants. In Inter-Cascade, the update loop is fundamentally different: the weak LLM decides when to update, and the strong LLM decides what to write, without any human in the loop. The stored content is not user utterances or QA pairs, but LLM-generated strategies and reasoning structures distilled from a stronger model. Rather than personalizing to a single user, Inter-Cascade uses interaction between two models to build a reusable strategic knowledge base for many users and tasks.

*Agentic RAG* A third, increasingly prominent direction combines RAG with multi-agent or agentic architectures (Li et al., 2025). In these systems, different agents are assigned distinct roles, e.g., planner, retriever, answer generator, or verifier. Those agents collaborate via tool calls and message passing. For centralized systems like MA-RAG (Nguyen et al., 2025b), HM-RAG (Liu et al., 2025), and SurgRaw (Low et al., 2025), the focus is on managing the workflow, such as deciding when to use the retriever to access the existing database. Decentralized methods like M-RAG (Wang et al., 2024c) and MDocAgent (Han et al., 2025) consider retrieval from partitioned databases. There are also works like RECOND (end-to-end generation) (Xu et al., 2025)Hippo (knowledge-graph) (Gutiérrez et al., 2025), IM-RAG (multi step refinement) (Yang et al., 2024) and FAIR-RAG (fair retrieval) (Shrestha et al., 2024) propose algorithms to refine answers from RAG database. However, in all such designs, the RAG component itself remains an external, fixed resource: agents coordinate how to use RAG, but no agent is responsible for constructing a new corpus of knowledge for others. Inter-Cascade differs from these agentic RAG systems in two key aspects. First, there are only two "agents": a weak LLM and a strong LLM, but their interaction is explicitly teacher–student and online knowledge distillation, rather than mere division of labor. Second, the strong LLM actively produces the knowledge store that the weak LLM later retrieves, making the RAG-like database a product of model interaction rather than a static tool.

Across all three categories, existing RAG approaches either (i) operate over a fixed, human-curated external corpus, (ii) update a memory store using human dialogue and feedback, or (iii) update a memory using self history for personalization without knowledge transfer. To our knowledge, Inter-Cascade is the first framework where a weak LLM and a strong LLM jointly and autonomously build a RAG-like corpus under the framework of LLM Cascade, with the weak model deciding when to consult and update it, and the strong model providing the organized knowledge. This yields a new form of online, interaction-driven distillation, particularly suitable for small models without access to large external knowledge bases or the Internet.

**Other related topics** There are also a weak model and strong model in *Speculative decoding* (Leviathan et al., 2023; Narasimhan et al., 2025). In speculative decoding, the weak model works as a answer draft while the strong model work as a verifier to speed up the generation compared to only using strong model. However, in Inter-Cascade, Strong LLM is called only when the Weak LLM is unable to handle current query. *CombLM* (Ormazabal et al., 2023) and *LLM Debate* (Irving et al., 2018; Du et al., 2023; Estornell & Liu, 2024; Khan et al., 2024; Zhou et al., 2025) are other branches of works that also involve interaction between LLMs. CombLM integrates the logit distribution of two LLMs while LLM Debate requires different LLMs to argue and refine their initial answers and eventually reach consensus through multiple rounds of interaction. The key difference between Inter-Cascade and them is that Inter-Cascade let the Strong LLM and Weak LLM work in a sequential order can conduct early stop to save tokens.

---

**Algorithm 3** Inter-Cascade Inference Pipeline

---

**Input:** Test set $\mathcal{T} = \{q_1, \ldots, q_I\} \subseteq \mathcal{Q}$; LLM $M_n$ with deferral function $d_n$, generation function $g_n$, strategy repository $\text{Repo}_n$ and strategy generator $h_n$.
    **Deferral convention:** $0 = $ handle locally, $1 = $ defer/forward.

1:   $\text{Repo} = \emptyset$
2: **for** $n \leftarrow 1$ **to** $N$ **do**
3:     **for** $i \leftarrow 1$ **to** $I$ **do**
4:        **if** $n < N$ **then**
5:           **(Strategy matching)**
6:           $[s_i^{t_1}, s_i^{t_2}, \ldots, s_i^{t_k}] \leftarrow f_n(q_i, \text{Repo}_n)$      ▷ Find most relevant top-$k$ strategies to $q_i$
7:           $q_i' \leftarrow [q_i, s_i^{t_1}, s_i^{t_2}, \ldots, s_i^{t_k}]$          ▷ concatenate query and strategies
8:        **else**
9:           $q_i' = q_i$           ▷ Last LLM doesn't maintain $\text{Repo}$

10:        **(Deferral Decision)**
11:        **if** $d_n(q_i') = 0$ **then**
12:           generate answer $a_i \leftarrow g_1(q_i')$        ▷ Answer locally at Weaker LLM
13:           $s_{\text{new}} \leftarrow h(q_i)$
14:           $\text{Repo}_{<n} \leftarrow \text{Repo}_{<n} \cup \{s_{\text{new}}\}$      ▷ Add strategy to all the weaker LLMs
15:        **else**
16:           **if** $n < N$ **then**
17:              Pass              ▷ Defer to next level
18:           **else**
19:              Discard current query $q_i$     ▷ None of LLMs are confident to answer the query

---

## C    ORDER OF LLMS

To distinguish two LLMs into strong model $M_s$ and weak model $M_w$, we make following definitions. For a task distribution $\mathcal{D}$, we denote the performance of a model $M$ by $\text{Perf}(M)$, which can be instantiated by measures such as the expected accuracy or negative loss on $\mathcal{D}$. Similarly, we let $\text{Cost}(M)$ represent the expected cost of using $M$ on $\mathcal{D}$, such as the price, latency, or required computation resource. Note that $\text{Cost}$ also depends on the task distribution $\mathcal{D}$, for simplicity, we only use the notation $\text{Cost}(M)$. We say that $M_w$ is weaker than $M_s$ if $\text{Perf}(m_w) \leq \text{Perf}(m_s)$, and that it is cheaper if $\text{Cost}(m_w) \leq \text{Cost}(m_s)$. To simplify notation, we introduce the shorthand relation

$$M_w \preccurlyeq_{\text{wbc}} M_s$$

if and only if

$$\text{Perf}(M_w) \leq \text{Perf}(M_s) \quad \text{and} \quad \text{Cost}(M_w) \leq \text{Cost}(M_s),$$

where the term "wbc" represents "weaker but cheaper". Consider a multi-LLM inference/generation system, which contains $N$ LLM models, $\mathcal{M} = \{M_1, M_2, \ldots, M_N\}$, with different capacities and use costs to a query. WLOG, we assume that $M_1 \preccurlyeq_{\text{wbc}} M_2 \preccurlyeq_{\text{wbc}} \ldots \preccurlyeq_{\text{wbc}} M_N$.

## D    ALGORITHM FOR GENERAL INTER-CASCADE

Since Inter-Cascade is scalable to any number of layers for LLM, the general Inter-Cascade pipeline for $N$-LLM cascade system is shown in Algo. 3.

## E    PROOF: CLOPPER-PERSON UPPER BOUND AS A BETA QUANTILE

In the lemma below, we apply the Clopper-Pearson upper bound to rewrite $R^+(\lambda)$, yielding a clearer form that facilitates computation. This helps the proof of Theorem 2.2 and Theorem G.1.

**Lemma E.1** (Clopper–Pearson upper bound as a Beta quantile). *Let $n(\lambda) \in \mathbb{N}$ be the number of evaluated items at threshold $\lambda$, let $R(\lambda) \in [0,1]$ denote the unknown risk, and suppose*

$$X \sim \mathrm{Bin}\big(n(\lambda),\, R(\lambda)\big),$$

*and $x \in \{0,1,\ldots,n(\lambda)\}$ is the number of error observed. Write $\widehat{R}(\lambda) = x/n(\lambda)$. For a fixed $\delta \in (0,1)$, define the one-sided $(1-\delta)$ upper confidence limit by*

$$\widehat{R}^+(\lambda) := \sup \Big\{ p \in [0,1] :\ \Pr_p\big(\mathrm{Bin}(n(\lambda),p) \le x\big) \ge \delta \Big\}.$$

*Then*

$$\widehat{R}^+(\lambda) = \mathrm{Beta}^{-1}\big(1-\delta;\, x+1,\, n(\lambda)-x\big)$$

*with the usual edge conventions $\mathrm{Beta}^{-1}(1-\delta; 1, n) = 1 - \delta^{1/n}$ when $x = 0$ and $\widehat{R}^+(\lambda) = 1$ when $x = n(\lambda)$.*

*Proof.* For fixed $x < n(\lambda)$ the map $p \mapsto F(p) := \Pr\big(\mathrm{Bin}(n(\lambda),p) \le x\big)$ is strictly decreasing in $p$, so the set in the definition of $\widehat{R}^+(\lambda)$ is an interval $[0, p^\star]$ and the supremum $p^\star$ uniquely solves

$$F(p^\star) = P\big(\mathrm{Bin}(n(\lambda),p^\star) \le x\big) = \delta. \tag{3}$$

Using the standard identity linking the binomial tail to the regularized incomplete beta function, for integers $0 \le x \le n(\lambda) - 1$,

$$P(X \le x) = \sum_{k=0}^{x} \binom{n(\lambda)}{k} p^k (1-p)^{n(\lambda)-k} = 1 - I_p\big(x+1,\, n(\lambda)-x\big),$$

where $I_p(a,b)$ is the CDF of $\mathrm{Beta}(a,b)$ at $p$. Plugging this into equation 3 gives

$$I_{p^\star}\big(x+1,\, n(\lambda)-x\big) = 1 - \delta,$$

so $p^\star$ is the $(1-\delta)$ quantile of the $\mathrm{Beta}\big(x+1,\, n(\lambda)-x\big)$ distribution:

$$p^\star = \mathrm{Beta}^{-1}\big(1-\delta;\, x+1,\, n(\lambda)-x\big).$$

This equals $\widehat{R}^+(\lambda)$ by definition. The stated edge cases follow from $F(p) = (1-p)^{n(\lambda)}$ when $x = 0$ and from monotonicity when $x = n(\lambda)$. $\qquad\square$

## F    PROOF: UNCHANGED THRESHOLD

**Theorem F.1.** *Suppose that $\widehat{R}^+(\lambda)$ is a monotonic decreasing function of $\lambda$. Fix $\delta \in (0,1)$ and an integer $n \ge 1$. For $x \in \{0,1,\ldots,n\}$, $\epsilon \in (0,1]$, and $b \in [1,\infty)$. Suppose that $\min\{\epsilon x + 1, n - \epsilon x\}$ is moderately large and $1 - \delta$ is not an extreme tail, then:*

*(a) Decrease in value. $\alpha(\epsilon,b) \le \alpha(1,1)$ when $\epsilon \in (0,1]$ and $b \in [1,\infty)$.*

*(b) Normal approximation for the amount of decrease. Let $z := \Phi^{-1}(1-\delta)$, where $\Phi$ is the Normal cumulative distribution function, when $n$ is large enough, the decrease of the risk under same level of tolerance is given by,*

$$\alpha(1,1) - \alpha(\epsilon,b) \approx \left( \frac{x+1}{n+1} - \frac{\epsilon x + 1}{bn+1} \right) + z \left[ \sqrt{\frac{(x+1)(n-x)}{(n+1)^2(n+2)}} - \sqrt{\frac{(\epsilon x+1)(bn-\epsilon x)}{(bn+1)^2(bn+2)}} \right].$$

*Proof.* We use a Beta function to represent the variable $\widehat{R}^+(\lambda)$, which is equivalent to the risk $\alpha$, when $\widehat{R}^+(\lambda)$ is a monotonic decreasing function of $\lambda$. We then use the approximation to Beta function to evaluate the decrease of $\alpha$ by definition. For the convenience of statement of our theories, we define that $\alpha(\epsilon, b)$ as the the value of risk bound $\alpha$ when the obtained $\lambda$ satisfies $n(\lambda) = bn$ and incorrectly answered queries among $n(\lambda)$ is $x(\lambda) = \epsilon x$, given the $\delta$ fixed. (a) Notice that we assume that $\widehat{R}^+(\lambda)$ is a monotonic decreasing function of $\lambda$. Let us suppose that $\lambda_0$ satisfies that $n(\lambda_0) = bn$ and $x(\lambda_0) = \epsilon x$. By Algorithm 1, this shows that $\widehat{R}^+(\lambda_0) = \alpha(\epsilon, b)$.

From Lemma E.1, we know that

$$\alpha(\epsilon, b) := \mathrm{Beta}^{-1}\big(1 - \delta;\ \epsilon x + 1,\ bn - \epsilon x\big).$$

Let $p_1 = \mathrm{Beta}^{-1}(1-\delta;\ x+1,\ n-x)$. Then, by the property of Beta distribution, $P\big(\mathrm{Bin}(n, p_1) \le x\big) = \delta$. It follows that,

$$P\big(\mathrm{Bin}(bn, p_1) \le \epsilon x\big) \ \le \ P\big(\mathrm{Bin}(n, p_1) \le x\big) = \delta,$$

because lowering the threshold ($\epsilon x \le bx$) and increasing trials ($bn \ge n$) makes the left tail event rarer. Let us assume that $p_2 = \mathrm{Beta}^{-1}(1 - \delta;\ \epsilon x + 1,\ bn - \epsilon x)$. From the proof of Lemma E.1, it is equivalent to that $P\big(\mathrm{Bin}(bn, p_2) \le \epsilon x\big) = \delta$. It follows that $P\big(\mathrm{Bin}(bn, p_2) \le \epsilon x\big) = \delta \ge P\big(\mathrm{Bin}(bn, p_1) \le \epsilon x\big)$, which implies that $p_2 \le p_1$. Hence the new upper bound $p_2 = \mathrm{Beta}^{-1}(1 - \delta;\ x' + 1,\ n' - x')$ satisfies $p_2 \le p_1$. This shows the statement (a).

(b) Write

$$\mu_{\epsilon,b} := \frac{\epsilon x + 1}{bn + 1}, \qquad \sigma_{\epsilon,b} := \sqrt{\frac{(\epsilon x + 1)(bn - \epsilon x)}{(bn + 1)^2(bn + 2)}}.$$

In the large–sample, interior regime, e.g., $\min\{\epsilon x + 1,\ n - \epsilon x\} \gg 1$ and $x/n$ bounded away from $0$ and $1$,

$$\mathrm{Beta}^{-1}\big(1 - \delta;\ \epsilon x + 1,\ bn - \epsilon x\big) \ = \ \mu_{\epsilon,b} \ + \ z\,\sigma_{\epsilon,b} \ + \ O\!\left(\frac{1}{n}\right).$$

This is by the approximation to Beta distribution by normal distribution. Calculate $\alpha(1, 1) - \alpha(\epsilon, b)$ demonstrate the result of theorem. $\square$

## G    PROOF: UNCHANGED USED QUERIES

Other than the case that the threshold remains unchanged, which is analyzed above, another case may be that when the user want the same number of queries to be covered by the Weak LLM during two rounds of queries (before and after adding strategies), one of which has a better Weak LLM. Such a case controls the cost. This case considers the influence of a better Weak LLM to our pipeline. In this case, we instead assume that $n(\lambda) = n(\lambda')$, and abbreviate them as $n$ for simplicity, which ensures the same coverage of Weak LLM. The number of wrongly answered queries before and after getting a better Weak LLM are denoted by $x$ and $\epsilon x$, and we still estimate the decrease of $\alpha$ under the same level of tolerance $\delta$. We give an approximation on the change rate of the risk bound with respect to the proportion of decrease of errors. We denote by $\alpha(\epsilon)$ the $\alpha(\epsilon, b = 1)$ for simplicity, and present the analysis in Theorem G.1.

**Theorem G.1.** *Suppose that $\widehat{R}^+(\lambda)$ is a monotonic decreasing function of $\lambda$. Fix $\delta \in (0, 1)$ and an integer $n \ge 1$. For $x \in \{0, 1, \ldots, n\}$ and $\epsilon \in (0, 1]$. Suppose that $\min\{\epsilon x + 1,\ n - \epsilon x\}$ is moderately large and $1 - \delta$ is not an extreme tail, then:*

*(a) Exact monotonicity.* $\alpha(\epsilon)$ *is strictly increasing in $\epsilon$. In particular, for any $\epsilon \in (0, 1)$,*

$$\alpha(\epsilon) \ < \ \alpha(1).$$

*(b) Normal approximation for the amount of decrease.* *Let $z := \Phi^{-1}(1 - \delta)$, for $\epsilon$ near $1$,*

$$\alpha(1) - \alpha(\epsilon) \approx (1 - \epsilon)\left[\frac{x}{n + 1} + \frac{z}{2(n + 1)\sqrt{n + 2}}\frac{x(n - 1 - 2x)}{\sqrt{(x + 1)(n - x)}}\right].$$

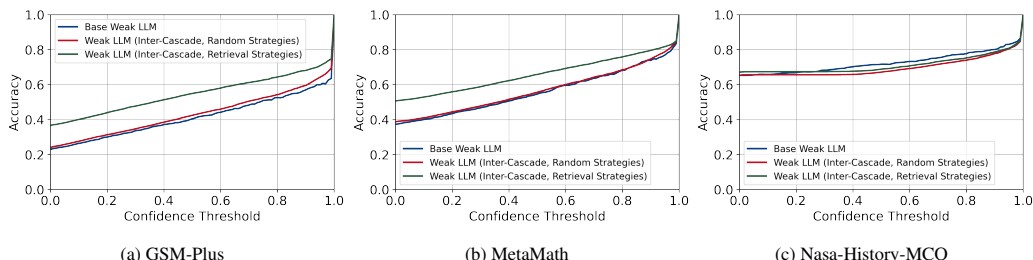

(a) GSM-Plus        (b) MetaMath        (c) Nasa-History-MCQ

Figure 3: Accuracy as a function of the confidence threshold for the base Weak LLM and for the Weak LLM within the Inter-Cascade using random and retrieval strategies across three benchmarks.

*Hence the decrease is approximately linear in $(1 - \epsilon)$ with the coefficient in brackets; in particular, when $x \leq n/2$ the variance term is nonnegative and the decrease is at least $(1 - \epsilon) x/(n + 1)$ to first order.*

*Proof.* (a) Similar to the proof of the statement (a) of Theorem 2.2, increasing $x$ moves mass to the right in the Binomial, so the lower-tail CDF in $p$ decreases and its $(1 - \delta)$ quantile increases; with $n$ fixed this is equivalent to $\alpha(\epsilon)$ being strictly increasing in $\epsilon$.

(b) Similar to the proof of the statement (a) of Theorem 2.2, notice that

$$\alpha(\epsilon, 1) := \mathrm{Beta}^{-1}\big(1 - \delta; \, \epsilon x + 1, \, n - \epsilon x\big).$$

For $i = \epsilon x + 1$, $j = n - \epsilon x$, the Beta $(i, j)$ mean and variance are $\mu_\epsilon = i/(i + j)$ and $\sigma_\epsilon^2 = ij/[(i + j)^2 (i + j + 1)]$. Approximating the $(1 - \delta)$ quantile by the Normal formula gives $\alpha(\epsilon) = \mu_\epsilon + z\sigma_\epsilon + O(1/n)$. Differentiate at $\epsilon = 1$ to obtain the first-order change:

$$\frac{d\mu_\epsilon}{d\epsilon}\bigg|_{\epsilon=1} = \frac{x}{n + 1}, \qquad \frac{d\sigma_\epsilon}{d\epsilon}\bigg|_{\epsilon=1} = \frac{1}{2(n + 1)\sqrt{n + 2}} \cdot \frac{(n - 1 - 2x)x}{\sqrt{(x + 1)(n - x)}}.$$

A first-order Taylor expansion around $\epsilon = 1$ yields the displayed approximation. $\qquad\square$

## H   CONFIDENCE DISTRIBUTION

Figures 3 and 4 present results for the GSM-Plus, MetaMath, and Nasa-History-MCQ datasets, complementing the GSM-Symbolic analyses in the main text.

Figure 3 shows accuracy as a function of the confidence threshold for the base Weak LLM and for the Weak LLM within the Inter-Cascade using random and retrieval strategies. For each threshold, only queries with confidence equal to or above the threshold are considered, and accuracy is calculated as the proportion of correct predictions. Across the reasoning datasets (GSM-Plus and MetaMath), the Inter-Cascade with retrieval strategies consistently improves accuracy over the baseline and random-strategy variants. For the factual non-reasoning dataset (Nasa-History-MCQ), the Inter-Cascade achieves comparable performance.

Figure 4 depicts the distribution of query confidence for the three benchmarks. Across all datasets, the Inter-Cascade with retrieval strategies concentrates probability mass near high confidence (0.9–1.0), whereas the base and random-strategy variants place more mass at lower confidence levels. These results further confirm that providing strategies helps the Weak LLM not only produce more accurate predictions but also better calibrate its confidence.

## I   FULL DESCRIPTION OF BENCHMARKS

**GSM-Symbolic.** The GSM-Symbolic benchmark, released by Apple's team (Mirzadeh et al., 2025), is a structured variant of GSM8K (Cobbe et al., 2021b). Unlike traditional benchmarks such as GSM8K, which present problems in a plain context, GSM-Symbolic reformulates problems into a

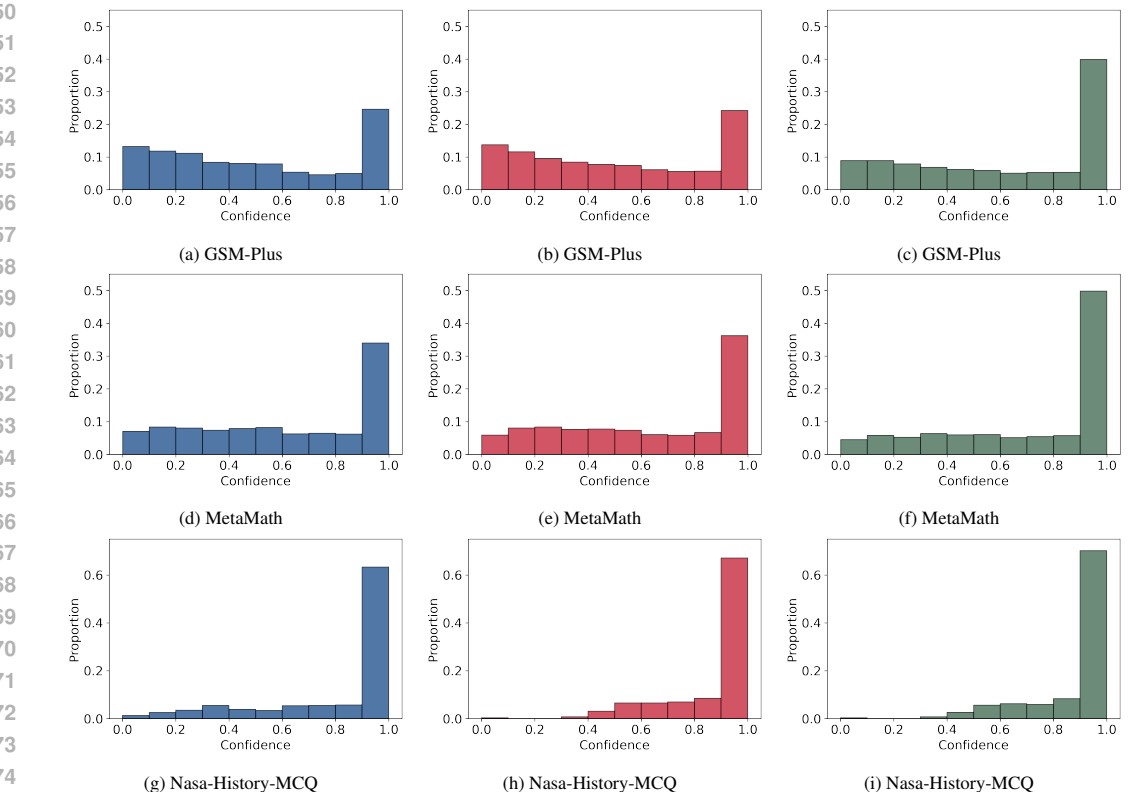

Figure 4: Confidence histograms for three benchmarks. Columns correspond to (a)(d)(g) the base Weak LLM, (b)(e)(h) the Weak LLM within the Inter-Cascade using random strategies, and (c)(f)(i) the Weak LLM within the Inter-Cascade using retrieval strategies. Across all datasets, the Inter-Cascade with retrieval strategies concentrates probability mass near high confidence (0.9–1.0), while the base and random-strategy variants place more mass at lower confidence levels.

more structured and abstract format following a symbolic template, providing a more reliable measure of models' reasoning capabilities. The dataset contains $12,500$ grade-school math problems. We randomly sample $1,250$ problems as the calibration set for threshold computation and use the remaining $11,250$ problems as the test set. The prompt template and an example problem are provided in Appendix L.

**GSM-Plus.** GSM-Plus (Li et al., 2024) is derived from the $1,319$ test questions in GSM8K by introducing eight types of question variations: numerical substitution, digit expansion, integer-decimal-fraction conversion, adding operation, reversing operation, problem understanding, distractor insertion, and critical thinking. GSM-Plus thus comprises a total of $10,552$ question variations. We randomly sample $1,048$ problems as the calibration set for threshold computation and use the remaining $9,504$ problems as the test set. The prompt template and an example problem are provided in Appendix L.

**MetaMath.** MetaMath (Yu et al., 2024) is a dataset generated by bootstrapping the mathematical benchmarks GSM8K (Cobbe et al., 2021b) and MATH (Hendrycks et al., 2021). The augmentation is performed in both forward and backward directions. In the forward direction, MetaMath contains the original and LLM-rephrased questions, while in the backward direction, it includes self-verification questions and FOBAR questions (Jiang et al., 2024), resulting in a total of $395K$ diverse problems. For our experiments, we randomly select $1,000$ problems as the calibration set for threshold computation and use $20,000$ additional problems as the test set. The prompt template and an example problem are provided in Appendix L.

**NASA-History-MCQ.** NASA-History-MCQ (Fleith, 2025) is a multiple-choice question benchmark on the history of NASA. It contains $7.47K$ questions, and each question provides four answer choices. We randomly sample $1,000$ problems as the calibration set for threshold computation and

use the remaining $6,469$ problems as the test set. The prompt template and an example problem are provided in Appendix L.

**BarExamQA.** BarExamQA (Zheng et al., 2025) is a legal reasoning benchmark constructed from real U.S. bar examination questions. Each question is posed in a multiple-choice format and requires multi-step legal reasoning over complex legal fact patterns. BarexamQA contains a total of $954$ problems, we randomly sample $95$ problems as the calibration set for threshold computation and remaining $859$ as the test set.

**BigBench Hard.** BIG-Bench Hard (Suzgun et al., 2022) is a subset of 23 particularly challenging BIG-Bench tasks for which no prior result from (Srivastava et al., 2022) has outperformed the average human-rater score. It is a diverse benchmark designed to test capabilities of language models on a diverse set of crowd-sourced tasks. The benchmark aims to focus on the problems that beyond the capabilities of existing LLMs. We use $5412$ problems as test set and $599$ problems as calibration set for threshold computation. The calibration set are selected from each tasks with the same proportion.

**GSM8K.** GSM8K (Cobbe et al., 2021b) is a widely used grade-school math word problem benchmark designed to evaluate multi-step numerical reasoning. The dataset contains $7473$ training questions and $1719$ test questions, with each problem requiring several arithmetic operations and logical reasoning steps to reach the final answer. Following standard practice, we use problems in calibration set for threshold computation and use the remaining problems as the test set.

**MedMCQA.** MedMCQA (Pal et al., 2022) is a large-scale multiple-choice question benchmark in the medical domain. It covers high-quality AIIMS and NEET PG entrance exam MCQs covering $2400$ healthcare topics and $21$ medical subjects. It contains over $194,000$ questions, each with four answer choices and a single correct answer. We randomly sample $2,000$ problems as the calibration set for threshold computation and use $8000$ additional problems as the test set.

## J   EXTENSIVE EXPERIMENT ON MORE BENCHMARKS

Although the Inter-Cascade diagram is motivated by the real-world scenarios that contain similar or repeated tasks, we also provide the result of our Inter-Cascade on extensive benchmarks that are more diverse and do not contain explicit sample variants: GSM8K (Cobbe et al., 2021a), BigBench Hard (Suzgun et al., 2022), BarExamQA (Zheng et al., 2025) and MedMCQA (Pal et al., 2022). The full description of those benchmarks are in Appendix I. We firstly test the accuracy of each single LLM on those benchmarks and the result is in Table 9.

**Inter-Cascade vs. Jung's LLM Cascade.** We evaluate our *Inter-Cascade* pipeline and Jung's method, as shown in Table 10. Our method outperforms Jung's, with a $0.18\% - 3.96\%$ increase in Pipeline Accuracy. The Strong LLM Call Rate is reduced on all benchmarks, with reductions ranging from $1.52\%$ to $16.14\%$. Compared with the results on GSM-Symbolic, GSM-Plus and Meta-Math benchmarks, the accuracy improvement is not that large, but the more important part is that our Inter-Cascade can still reach a better trade-off between accuracy and cost since our method still remarkably reduce the usage of Strong LLM. These results indicate that *Inter-Cascade* pipeline is also beneficial across different categories of tasks on diverse benchmarks.

**Impact of Inter-Cascade on Weak LLM.** Having examined the overall pipeline improvements, including Pipeline Accuracy and Strong LLM Call Rate reduction, we now investigate how our proposed *Inter-Cascade* affects the Weak LLM. As shown in Table 11, our Weak LLM still outperforms the Weak LLM in the other pipeline across all benchmarks. The improvements on *Weak Accuracy* are between $0.91\%$ and $9.56\%$ and the improvements on *Weak Correct Accepted* are between $2.24\%$ and $15.56\%$. The results implies that even though we test our Inter-Cascade on diverse benchmarks, retrieving most similar problems and solution strategies can still help boosting the performance and confidence of Weak LLM.

According to experiment results for extensive benchmarks, it shows that Inter-Cascade not only work for tasks that contain constructive similarity, but also help in more general and diverse cases, since explicit or implicit similarity occurs everywhere and the pipeline in our Inter-Cascade take the advantage of the similarity nature of daily tasks.

**Token and API Cost Savings.** The results of analysis on cost and latency for extensive benchmarks are attached in Table 12 and Table 13. The tendency is similar: integrating with strategies, the token usages on Weak LLM increase between $115.89\%$ and $216.37\%$, but since the *Strong Call* decrease on all benchmark, the token usages on Strong LLM decrease between $1.28\%$ and $83.17\%$ and therefore, we can save $2.33\% \text{-} 83.94\%$ money on API price. On the other hand, the average latency change on each query is between $0.005\,\text{s}$ and $0.374\,\text{s}$ on different benchmarks, which is acceptable to the user experience.

Table 9: Accuracies of the base LLMs on extensive benchmarks

| Dataset | LLM | Accuracy | Dataset | LLM | Accuracy |
|---|---|---|---|---|---|
| GSM8K | gpt-3.5-turbo | 31.46% | BigBench | gpt-3.5-turbo | 49.75% |
| | gemini-2.0-flash | 74.83% | | gemini-2.0-flash | 78.80% |
| BarExamQA | gpt-3.5-turbo | 48.42% | MedMCQA | gpt-3.5-turbo | 62.80% |
| | gemini-2.0-flash | 78.95% | | gemini-2.0-flash | 83.05% |

## K EXTRA ABLATION STUDY

To better evaluate the performance and generalization capacity of Inter-Cascade, we set up extra ablation studies in this section.

### K.1 COLD START

To evaluate the effect of cold start of our strategy repository, we measure the dynamic pipeline accuracy for both Jung's method and our standard Inter-Cascade on GSM-Symbolic. The result in Figure 5 shows that at early stage, the pipeline accuracy for our Inter-Cascade is much close to baseline method: Jung (Jung et al., 2025). However, as the size of stored strategies increase, the performance of Inter-Cascade increase and gradually exceed Jung's method and eventually converges.

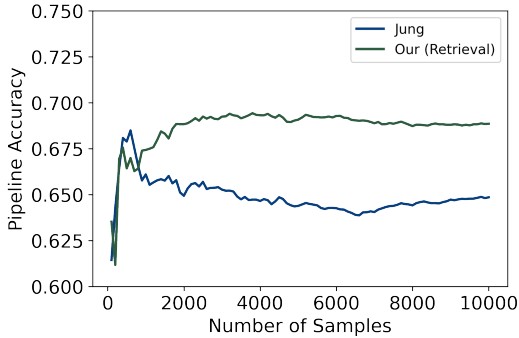

Figure 5: The dynamic of pipeline accuracy for both Jung's method and our standard Inter-Cascade on GSM-Symbolic.

### K.2 EFFECT OF STRATEGIES NUMBER

To evaluate the effect the number of strategies we matched for each queries, we test the pipeline accuracy with different number of strategies that used for integrating with the input of Weak LLM. The result in Figure 6 shows that the trend of pipeline accuracy is increasing first, reaching peak and then decreasing along with the number of strategies. The result makes sense because too few strategies might not retrieve the best strategy in repository, while too many strategies might distract the answer from certain query question, furthermore, there is a chance that the longer contexts may exceed the the maximum limit of the input context window. Both factors might undermine the performance of the pipeline accuracy. In our experiment on GSM-Symbolic benchmark, the empirical best number of strategies $k$ is 2.

Table 10: Results across extensive datasets using different pipelines. "Jung" denotes Jung's LLM-Cascade and "Our (Retrieval)" denotes the Inter-Cascade with similarity-based retrieval. The number of strategies is fixed at $k = 2$ for both Inter-Cascade settings. Metrics reported are Pipeline Accuracy (Pipeline Acc.), Strong LLM Call Rate (Strong Call), and Coverage Rate (Cov.). (a) GSM8K: For the Strong LLM, $\alpha_s = 0.2, \delta_s = 0.8, \lambda_s = 0.44$. For the Weak LLM, $\alpha_w = 0.5, \delta_w = 0.5, \lambda_w = 0.49$. (b) BigBench: No threshold is applied for the Strong LLM. For the Weak LLM, $\alpha_w = 0.4, \delta_w = 0.6, \lambda_w = 0.61$. (c) BarExamQA: No threshold is applied for the Strong LLM. For the Weak LLM, $\alpha_w = 0.5, \delta_w = 0.5, \lambda_w = 0.51$. (d) MedMCQA: No threshold is applied for the Strong LLM. For the Weak LLM, $\alpha_w = 0.3, \delta_w = 0.8, \lambda_w = 0.69$.

| Data | Pipeline | Pipeline Acc. (%)↑ | Strong Call (%)↓ | Cov. (%) |
|------|----------|-------------------|------------------|----------|
| GSM8K | Jung | 59.02 | 37.03 | 95.95 |
| | Our (Retrieval) | **60.62** | **35.46** | **96.05** |
| BigBench | Jung | 64.14 | 33.04 | 100.00 |
| | Our (Retrieval) | **64.32** | **23.84** | 100.00 |
| BarExamQA | Jung | 57.39 | 23.17 | 100.00 |
| | Our (Retrieval) | **58.67** | **21.65** | 100.00 |
| MedMCQA | Jung | 71.69 | 18.74 | 100.00 |
| | Our (Retrieval) | **75.65** | **2.60** | 100.00 |

Table 11: Results on Weak LLM across extensive datasets. Reported metrics are Weak LLM Accuracy (Weak Acc.) and Weak Correct Accepted (Weak Corr. Accpt.). Parameter settings are the same as in Table 10.

| Data | Pipeline | Weak Acc. (%)↑ | Weak Corr. Accpt. (%)↑ |
|------|----------|----------------|------------------------|
| GSM8K | Jung | 37.06 | 33.38 |
| | Our (Retrieval) | **39.30** | **35.62** |
| BigBench | Jung | 49.02 | 39.34 |
| | Our (Retrieval) | **49.93** | **46.60** |
| BarExamQA | Jung | 47.50 | 39.81 |
| | Our (Retrieval) | **51.22** | **43.31** |
| MedMCQA | Jung | 64.95 | 58.16 |
| | Our (Retrieval) | **74.51** | **73.72** |

## K.3 RESULTS ON NEW LLM PAIRS

To show that our Inter-Cascade is a framework that work general multiple LLM collaboration systems, we also test the result on different choice of Weak LLM and Strong LLM. We switch our Weak LLM to Gemini-2.0-flash and switch our Strong LLM to Gemini-2.5-flash. The results on single LLM are in Table 14. We also analyze the performance on those metrics: Pipeline Accuracy, Strong Call Rate, Weak Accuracy and Weak Correct Accept in Table 15 and Table 16. The results shows that although we test on different pairs of Weak LLM and Strong LLM, the trend doesn't change: Inter-Cascade would help improve the accuracy of Weak LLM, pipeline accuracy, reduce the the usage of Strong LLM, reaching a better trade-off between the Accuracy and Cost in LLM Cascade systems.

Table 12: Token and API cost changes across extensive datasets for Inter-Cascade compared with Jung's pipeline.

| Benchmark | Weak LLM Tokens | | | Strong LLM Tokens | | | Token Price |
|-----------|-------|-------|--------|-------|-------|--------|-------|
| | Total | Input | Output | Total | Input | Output | |
| GSM8K | +115.89% | +116.56% | -2.27% | -3.25% | -4.10% | -1.28% | -2.33% |
| BigBench | +134.53% | +135.32% | -5.47% | -26.37% | -30.90% | -19.67% | -22.70% |
| BarExamQA | +216.37% | +216.90% | +0.12% | -5.70% | -5.39% | -6.28% | -5.98% |
| MedMCQA | +129.64% | +130.70% | -0.16% | -84.74% | -85.58% | -83.17% | -83.94% |

Table 13: Processing Latency and Strategy Repository Size across extensive datasets. Retrieval refers to the time spent on strategies matching and ranking. Generation refers to time spent on generating answer via API.

| Benchmark | Tested Samples | Our | | | Jung | Repository Size |
|---|---|---|---|---|---|---|
| | | Total | Retrieval | Generation | Total | |
| GSM8K | 7473 | 1.344s | 0.005s | 1.339s | 1.216s | 6.3MB |
| BigBench | 5412 | 1.456s | 0.004s | 1.452s | 1.227s | 3.4MB |
| BarExamQA | 859 | 1.686s | 0.254s | 1.432s | 1.312s | 1.1MB |
| MedMCQA | 8000 | 0.975s | 0.004s | 0.971s | 0.970s | 6.3MB |

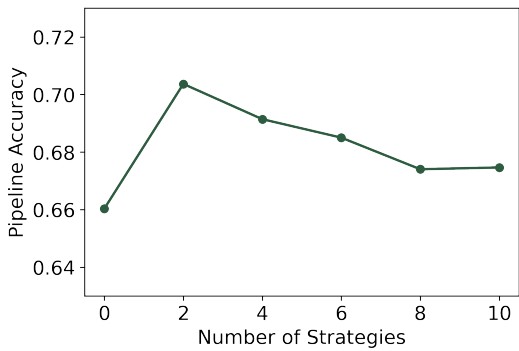

Figure 6: Effect of number of Strategies on pipeline accuracy for GSM-Symbolic Benchmark

## L  PROMPT TEMPLATES AND EXAMPLES

Table 17 and Table 18 present the strategy-free prompt templates for the four datasets, along with one example question per dataset. Table 19 - Table 22 show the strategy-based prompt templates and example inputs for each dataset. In our experiments, the number of strategies is set to $k = 2$; these strategies and their corresponding answers are generated by the Strong LLM. Since the pipeline operates without human intervention, all strategies that exceed the Strong LLM confidence threshold $\lambda_s$ are accepted. Consequently, the Repo may contain incorrect strategies or answers. Nonetheless, the results in Table 3 and Table 4 demonstrate the effectiveness of $\lambda_s$ and the robustness of our proposed Inter-Cascade pipeline.

Table 14: Accuracies of new pair of base LLMs on GSM-Symbolic Benchmark

| Dataset | LLM | Accuracy |
|---|---|---|
| GSM-Symbolic | gemini-2.0-flash | 69.36% |
| | gemini-2.5-flash | 89.28% |

Table 15: New LLM Pairs (Weak LLM: Gemini-2.0-flash; Strong LLM: Gemini-2.5-flash) Results on GSM-Symbolic dataset using different pipelines. "Jung" denotes Jung's LLM-Cascade and "Our (Retrieval)" denotes the Inter-Cascade with similarity-based retrieval. The number of strategies is fixed at $k = 2$ for both Inter-Cascade settings. Metrics reported are Pipeline Accuracy (Pipeline Acc.), Strong LLM Call Rate (Strong Call), and Coverage Rate (Cov.). GSM-Symbolic: No threshold is applied for the Strong LLM. For the Weak LLM, $\alpha_w = 0.2, \delta_w = 0.8, \lambda_w = 0.47$.

| Data | Pipeline | Pipeline Acc. (%)↑ | Strong Call (%)↓ | Cov. (%) |
|---|---|---|---|---|
| GSM-Symbolic | Jung | 79.10 | 19.10 | 100.00 |
| | Our (Retrieval) | **85.50** | **9.90** | 100.00 |

Table 16: New LLM Pairs (Weak LLM: Gemini-2.0-flash; Strong LLM: Gemini-2.5-flash) Results on Weak LLM across GSM-Symbolic dataset. Reported metrics are Weak LLM Accuracy (Weak Acc.) and Weak Correct Accepted (Weak Corr. Accpt.). Parameter settings are the same as in Table 15.

| Data | Pipeline | Weak Acc. (%)↑ | Weak Corr. Accpt. (%)↑ |
|---|---|---|---|
| GSM-Symbolic | Jung | 64.20 | 63.40 |
| | Our (Retrieval) | **77.00** | **76.80** |

Table 17: Strategy-free prompt template with example questions from GSM-Symbolic, GSM-Plus, and Meta-Math

---

**Prompt Template:**

Based on the question below, please strictly follow this format when answering:
1. Start with [Strategy] section explaining the general approach for solving similar problems;
2. End with [Answer] section containing ONLY the value. (1) Do NOT include units such as minutes, feet, etc.; (2) If the question asks for a percentage, ONLY provide the number (e.g., answer $20$ instead of $20\%$ ); (3) Do NOT include any explanations; (4) If there is no answer, RETURN None as the value.

Example:
[Question]: $x + y = 10, y = 4$, what is $x$?
[Strategy]: To solve for $x$, isolate $x$ by subtracting $y$ from both sides of the equation. $x = 10 - y = 10 - 4 = 6$.
[Answer]: $6$

Now answer this question:
[Question]: {question}
[Strategy]:
[Answer]:

---

**GSM-Symbolic Example Question:**

[Question]: A fog bank rolls in from the ocean to cover a city. It takes $495$ minutes to cover every $95$ miles of the city. If the city is $95$ miles across from the oceanfront to the opposite inland edge, how many minutes will it take for the fog bank to cover the whole city?

**GSM-Plus Example Question:**

[Question]: A clothing store has $60$ white shirts and $70$ floral shirts. Two-thirds of the white shirts have collars, and $30$ of the floral shirts have buttons. How many more floral shirts with no buttons are there than white shirts with no collars?

**MetaMath Example Question:**

[Question]: Lara bought $52$ stems of flowers at the flower shop. She gave $15$ flowers to her mom and gave $x$ more flowers than she gave to her mom to her grandma. She put the rest in a vase. How many stems of flowers did Lara put in the vase? If we know the answer to the above question is $16$, what is the value of unknown variable $x$?

Table 18: Strategy-free prompt template with example question from NASA-History-MCQ

**Prompt Template:**

Based on the question below, please strictly follow this format when answering:
1. Start with [Strategy] section explaining the general approach for solving similar problems;
2. End with [Answer] section containing ONLY the single capital letter of the correct option (exactly one of: A, B, C, D). Do NOT include the option text or any explanation.

Example:
[Question]: Which of the following was a direct result of the Treaty of Versailles (1919)?
A. The outbreak of World War II
B. The United States declaring war on Germany and entering World War I
C. The establishment of the Fascist regime in Italy
D. The creation of the League of Nations
[Strategy]: Recall that the treaty ended World War I and included peacekeeping measures. Compare options with the treaty: only the League of Nations was directly established by it.
[Answer]: D

Now answer this question:
[Question]: {question}
[Strategy]:
[Answer]:

---

**Nasa-History-MCQ Example Question:**

[Question]: What three distinct stages comprise current psychological support protocols for US astronauts?
A. Individual, group, family
B. Training, mission, debriefing
C. Cognitive, emotional, behavioral
D. Preflight, in-flight, postflight

Table 19: Strategy-based prompt template with example input from GSM-Symbolic

**Prompt Template:**

Using the following problem examples:
{strategy}

Based on the question below, please strictly follow this format when answering:
1. Start with [Strategy] section explaining the general approach for solving similar problems;
2. End with [Answer] section containing ONLY the value. (1) Do NOT include units such as minutes, feet, etc.; (2) If the question asks for a percentage, ONLY provide the number (e.g., answer 20 instead of 20%); (3) Do NOT include any explanations; (4) If there is no answer, RETURN None as the value.

Now answer this question:
[Question]: {question}
[Strategy]:
[Answer]:

---

**GSM-Symbolic Example Input:**

Using the following problem examples:
1. [Question]: A fog bank rolls in from the ocean to cover a city. It takes 93 minutes to cover every 2 miles of the city. If the city is 24 miles across from the oceanfront to the opposite inland edge, how many minutes will it take for the fog bank to cover the whole city?
[Strategy]: First, determine how many 2-mile segments are in the 24-mile city. Divide the total distance of the city by the length of each segment. Then, multiply the number of segments by the time it takes to cover each segment to find the total time.
[Answer]: 1116

2. [Question]: A fog bank rolls in from the ocean to cover a city. It takes 114 minutes to cover every 3 miles of the city. If the city is 99 miles across from the oceanfront to the opposite inland edge, how many minutes will it take for the fog bank to cover the whole city?
[Strategy]: First, determine the rate at which the fog bank covers the city in miles per minute. Then, multiply this rate by the total distance of the city to find the total time it takes to cover the city. The rate is 3 miles / 114 minutes = $1/38$ miles per minute. The total time is ($1/38$ miles/minute) $*$ 99 miles = $99/38$ minutes. Simplify the fraction $99/38 = 2.60526315789$. Multiply 114 by $99/3$ to get the answer $114 * (99/3) = 114 * 33 = 3762$.
[Answer]: 3762

Based on the question below, please strictly follow this format when answering:
1. Start with [Strategy] section explaining the general approach for solving similar problems;
2. End with [Answer] section containing ONLY the value. (1) Do NOT include units such as minutes, feet, etc.; (2) If the question asks for a percentage, ONLY provide the number (e.g., answer 20 instead of 20%); (3) Do NOT include any explanations; (4) If there is no answer, RETURN None as the value.

Now answer this question:
[Question]: A fog bank rolls in from the ocean to cover a city. It takes 495 minutes to cover every 95 miles of the city. If the city is 95 miles across from the oceanfront to the opposite inland edge, how many minutes will it take for the fog bank to cover the whole city?
[Strategy]:
[Answer]:

Table 20: Strategy-based prompt template with example input from GSM-Plus

**Prompt Template:**

Using the following problem examples:
{strategy}

Based on the question below, please strictly follow this format when answering:
1. Start with [Strategy] section explaining the general approach for solving similar problems;
2. End with [Answer] section containing ONLY the value. (1) Do NOT include units such as minutes, feet, etc.; (2) If the question asks for a percentage, ONLY provide the number (e.g., answer $20$ instead of $20\%$ ); (3) Do NOT include any explanations; (4) If there is no answer, RETURN None as the value.

Now answer this question:
[Question]: {question}
[Strategy]:
[Answer]:

---

**GSM-Plus Example Input:**

Using the following problem examples:
1. [Question]: A clothing store has some white shirts and $50$ floral shirts. Half of the white shirts have collars, and $20$ of the floral shirts have buttons. How many more floral shirts with no buttons are there than white shirts with no collars?
[Strategy]: Let $W$ be the number of white shirts. The number of white shirts with collars is $W/2$, so the number of white shirts without collars is $W - W/2 = W/2$. The number of floral shirts is $50$. The number of floral shirts with buttons is $20$, so the number of floral shirts without buttons is $50 - 20 = 30$. We want to find the difference between the number of floral shirts without buttons and the number of white shirts without collars, which is $30 - W/2$. However, we are not given the number of white shirts, $W$. Therefore, we cannot determine the exact numerical value of the difference.
[Answer]: None

2. [Question]: A clothing store has $40$ white shirts and $50$ floral shirts. Half of the white shirts have collars, and some of the floral shirts have buttons. There are $10$ more floral shirts with no buttons than white shirts with no collars. How many floral shirts have buttons?
[Strategy]: 1. Calculate the number of white shirts with collars: $40/2 = 20$. 2. Calculate the number of white shirts without collars: $40 - 20 = 20$. 3. Calculate the number of floral shirts without buttons: $20 + 10 = 30$. 4. Calculate the number of floral shirts with buttons: $50 - 30 = 20$.
[Answer]: $20$

Based on the question below, please strictly follow this format when answering:
1. Start with [Strategy] section explaining the general approach for solving similar problems;
2. End with [Answer] section containing ONLY the value. (1) Do NOT include units such as minutes, feet, etc.; (2) If the question asks for a percentage, ONLY provide the number (e.g., answer $20$ instead of $20\%$ ); (3) Do NOT include any explanations; (4) If there is no answer, RETURN None as the value.

Now answer this question:
[Question]: A clothing store has $60$ white shirts and $70$ floral shirts. Two-thirds of the white shirts have collars, and $30$ of the floral shirts have buttons. How many more floral shirts with no buttons are there than white shirts with no collars?
[Strategy]:
[Answer]:

Table 21: Strategy-based prompt template with example input from MetaMath

---

**Prompt Template:**

Using the following problem examples:
{strategy}

Based on the question below, please strictly follow this format when answering:
1. Start with [Strategy] section explaining the general approach for solving similar problems;
2. End with [Answer] section containing ONLY the value. (1) Do NOT include units such as minutes, feet, etc.; (2) If the question asks for a percentage, ONLY provide the number (e.g., answer $20$ instead of $20\%$ ); (3) Do NOT include any explanations; (4) If there is no answer, RETURN None as the value.

Now answer this question:
[Question]: {question}
[Strategy]:
[Answer]:

---

**MetaMath Example Input:**

Using the following problem examples:
1. [Question]: Martha has $x$ crayons. She lost half of them, so she bought a new set of $20$ crayons. Martha has $29$ crayons in total after the purchase. What is the value of unknown variable $x$?
[Strategy]: Let $x$ be the number of crayons Martha initially had. She lost half of them, so she had $x/2$ crayons left. Then she bought 20 new crayons, so she had $x/2 + 20$ crayons. We are given that she has $29$ crayons in total, so $x/2 + 20 = 29$. Subtracting $20$ from both sides gives $x/2 = 9$. Multiplying both sides by $2$ gives $x = 18$.
[Answer]: $18$

2. [Question]: Justin and Sabrina at Pine Grove Elementary have $50$ pencils combined. Justin has $x$ more than twice as many pencils as Sabrina. Sabrina has $14$ pencils. What is the value of unknown variable $x$?
[Strategy]: Let $J$ be the number of pencils Justin has and $S$ be the number of pencils Sabrina has. We are given that $J + S = 50$ and $S = 14$. We are also given that Justin has $x$ more than twice as many pencils as Sabrina, which can be written as $J = 2S + x$. We can substitute $S = 14$ into the first equation to find $J$: $J + 14 = 50$, so $J = 50 - 14 = 36$. Now we can substitute $J = 36$ and $S = 14$ into the second equation: $36 = 2(14) + x$, so $36 = 28 + x$. Solving for $x$, we get $x = 36 - 28 = 8$.
[Answer]: $8$

Based on the question below, please strictly follow this format when answering:
1. Start with [Strategy] section explaining the general approach for solving similar problems;
2. End with [Answer] section containing ONLY the value. (1) Do NOT include units such as minutes, feet, etc.; (2) If the question asks for a percentage, ONLY provide the number (e.g., answer $20$ instead of $20\%$ ); (3) Do NOT include any explanations; (4) If there is no answer, RETURN None as the value.

Now answer this question:
[Question]: Lara bought $52$ stems of flowers at the flower shop. She gave $15$ flowers to her mom and gave $x$ more flowers than she gave to her mom to her grandma. She put the rest in a vase. How many stems of flowers did Lara put in the vase? If we know the answer to the above question is $16$, what is the value of unknown variable $x$?
[Strategy]:
[Answer]:

---

Table 22: Strategy-based prompt template with example input from NASA-History-MCQ

**Prompt Template:**

Using the following problem examples:
{strategy}

Based on the question below, please strictly follow this format when answering:
1. Start with [Strategy] section explaining the general approach for solving similar problems;
2. End with [Answer] section containing ONLY the single capital letter of the correct option (exactly one of: A, B, C, D). Do NOT include the option text or any explanation.

Now answer this question:
[Question]: {question}
[Strategy]:
[Answer]:

---

**Nasa-History-MCQ Example Input:**

Using the following problem examples:
1. [Question]: Beyond communication and care packages, what specific types of hardware or software aid psychological well-being during long-duration spaceflights?
A. Specialized dietary supplements to combat isolation
B. Automated exercise routines tailored to reduce stress
C. Psychological support hardware and software
D. Advanced life support systems with mood stabilizers
[Strategy]: The question asks about specific hardware or software that aids psychological well-being during long-duration spaceflights, beyond communication and care packages. We need to evaluate each option to see if it fits this description. Option A focuses on dietary supplements, which are not hardware or software. Option B describes automated exercise routines, which could involve software and hardware. Option C is too general, simply restating the question. Option D focuses on life support systems with mood stabilizers, which are not necessarily hardware or software designed specifically for psychological well-being. Therefore, option B is the most specific and relevant answer.
[Answer]: B

2. [Question]: What is the anticipated effect of constraints inherent in lunar and Martian missions on psychological support approaches?
A. Greater emphasis on real-time communication with Earth-based support teams
B. Increased reliance on virtual reality and AI companionship to mitigate isolation
C. A shift towards highly individualized psychological profiles and tailored interventions
D. A return to the mindset and strategies of earlier explorers and their families
[Strategy]: The question asks about the impact of constraints in lunar and Martian missions on psychological support. These constraints include isolation, limited resources, communication delays, and the need for self-sufficiency. Considering these limitations, the most likely effect would be a greater reliance on technologies that can provide support in the absence of immediate Earth-based assistance and a need for personalized approaches due to the unique challenges faced by each astronaut. Options A and D are less likely because of communication delays and the differences between modern space missions and earlier explorations. Option B is plausible, but option C is more comprehensive as it addresses the need for personalized support, which is crucial given the constraints.
[Answer]: C

Based on the question below, please strictly follow this format when answering:
1. Start with [Strategy] section explaining the general approach for solving similar problems;
2. End with [Answer] section containing ONLY the single capital letter of the correct option (exactly one of: A, B, C, D). Do NOT include the option text or any explanation.

Now answer this question:
[Question]: What three distinct stages comprise current psychological support protocols for US astronauts?
A. Individual, group, family
B. Training, mission, debriefing
C. Cognitive, emotional, behavioral
D. Preflight, in-flight, postflight
[Strategy]:
[Answer]:

