# OpenReview forum: "Not only a helper, but also a teacher: Interactive LLM Cascade"
_ICLR.cc/2026/Conference — Submitted to ICLR 2026_

### Official Review · Reviewer_qpUq · 2025-10-30

**Soundness:** 3
**Presentation:** 3
**Contribution:** 3
**Rating:** 6
**Confidence:** 4

**Summary:**

This paper presents a new LLM Cascade method. The proposed method involves adding (top-k related) historical question-strategy-answer information to the prompt when querying a weak model, using the in-context learning methodology.

**Strengths:**

1) I believe the LLM Cascade method proposed in this paper will be very simple and effective. I find this idea interesting.

2) Secondly, the overall presentation of the paper and its figures are clear, especially in the Section 2 (although I suggest the authors separate all the theoretical derivations into a dedicated subsection).

**Weaknesses:**

**Weaknesses and Suggestions**

1) For the experiment.
(i) Many cascade methods could be used for comparison, but they are not included in the experiments. The comparison with baselines is significantly lacking. (ii) Additionally, the performance of the weak and strong models used separately should also be reported.    Some other concerns: Another issue is why more well-known datasets, such as MATH and GSM8K, were not used.

2) For the method.
The authors overlook an important step: the introduction of the "strategy matching function", involving how multiple similarities are calculated to find the TOP-K samples.

2) For the related work.
(i) I suggest that the authors add a brief presentation of the most core related works in the main text. (ii) Additionally, it is recommended that the authors provide a categorized analysis of related works, rather than presenting them in a continuous flow (the authors may refer to the survey "Harnessing Multiple Large Language Models: A Survey on LLM Ensemble, arXiv 2025" for the classification of cascade methods). (iii) Furthermore, it would be beneficial to include an introduction to related works on "in-context learning involving the retrieval of the most relevant samples".

3) There are some unclear points and flaws in the abstract and introduction sections (though overall, the paper is easy to follow).
(i) The description of existing cascade methods in the introduction is not sufficiently clear and is presented from a high-level perspective. The statement "the deferral decision is trained offline" in the abstract is not entirely accurate, as some cascade methods do not require training.
(ii) The meaning and significance of "offline and online" for the proposed method are not clearly explained in the abstract and introduction, at least not in the abstract.
(iii) The explanation of the proposed method (in the second, third, and fourth paragraphs of the introduction) is not clear enough.
I suggest that the authors provide a case in the introduction or method section to explain the proposed method, including the use of the in-context learning-based prompt. This will help readers understand the method quickly.

**Questions:**

Please refer to the above "Weaknesses and Suggestions".

---

> ### Author Response · Authors · 2025-11-22
> **Response to Weaknesses:**
>
> We appreciate the reviewer’s insightful observations and constructive criticism.
>
> 1. **Choice of Baseline**:
>
>     - (i) The baseline we choose [1] is the first paper in LLM Cascade which provides theoretical guarantee on the error rate. Their work reaches SOTA compared to predictive probability[2] (which is widely used but easy to overestimate), verbalized confidence[3]. In our experiment, we also add ablations settings which contains two variants of the LLM Cascades.  Last but not least, our inter-Cascade can be integrated to any LLM Cascade, we choose the SOTA as the reference.
>
>     - (ii) The performance of sole LLM is in Table 1.
>
>     - (iii) We are adding experiments on the benchmarks that reviewers mentioned. We will update the result once finished.
>
>     [1] Jung, Jaehun, Faeze Brahman, and Yejin Choi. "Trust or escalate: Llm judges with provable guarantees for human agreement." arXiv preprint arXiv:2407.18370 (2024).
>
>     [2] Chen, Lingjiao, Matei Zaharia, and James Zou. "Frugalgpt: How to use large language models while reducing cost and improving performance." arXiv preprint arXiv:2305.05176(2023).
>
>     [3] Tian, Katherine, et al. "Just ask for calibration: Strategies for eliciting calibrated confidence scores from language models fine-tuned with human feedback." arXiv preprint arXiv:2305.14975 (2023).
>
> 2. **Strategy Matching Function**: Our framework is general to any similarity matching functions. The empirical method we used is listed in Section 3.2 Experiments settings. In our experience, We employ the all-MiniLM-L6-v2 transformer to produce 384-dimensional sentence embeddings and calculate cosine distances between the question embedding vector and each strategy-related embedding vector. Rank them with the distance and return top-$k$ vectors, together with corresponding strategy texts.
>
>  3. **Refinement on Related Work**:
>
>     - (i) Thanks for the suggestions, we enlarged the related work part, and added more details in the main text.
>
>     - (ii) We emphasized the categories of the related work, e.g., In LLM Cascade part, we address that LLM Cascade falls into the category of  Ensemble after inference LLM Ensemble. In Retrieval-Augmented Generation (RAG) part, we split the related work into three groups and made comparisons to each sub-category.
>
>     - (iii) As multiple reviewers mentioned, we group the "in-context learning involving the retrieval of the most relevant samples, dynamic RAG as History-aware RAG and made comparisons with our Inter-Cascade. We have tried our best but with the limit of pages, we just add brief outlines of related work in Introductions. The extensive comparison is placed in the appendix.
>
> 4. **Refinement on writing and presentation**
>
>     - (i) To be more rigorous, we rewrite that sentence as”the deferral decision rule is trained or derived by algorithms offline.” to conclude the deferral functions that are trained from.
>
>     - (ii) Here offline means the model training stage, online means the inference stage. We mentioned online because we want to emphasize that our method doesn't require training and can conduct during inference/generation of answers.
>
>     - (iii) we modified and added more supporting materials (motivations on the consideration of similar problems; related works.) to the abstract and introduction to try to describe the problem and our ideas clearer.  We also add clarification on motivations and applications in **common response 1 & 2**.

---

> > ### Author Response · Authors · 2025-12-03
> > **Additional Experiment Uploaded**
> >
> > We have added those experiments that you suggested, please also refer to the **point 8 and 9** in common response:
> >
> > 1. Continued Response to **1.Choice of Baseline**(in first round): we added experiments on four more benchmarks including GSM8K as reviewer mentioned. The results are shown in Table 10-13.
> >
> > **Lasted version of paper(v2) has been uploaded.**
> >
> > We deeply appreciate the time and effort you’ve dedicated to helping us refine and enhance our work!

---

### Official Review · Reviewer_jo4V · 2025-10-31

**Soundness:** 3
**Presentation:** 3
**Contribution:** 2
**Rating:** 4
**Confidence:** 3

**Summary:**

This paper introduces Inter-Cascade, an online and interactive extension of the LLM Cascade paradigm. Unlike standard cascades—which statically defer uncertain queries from a weak (cheap) model to a strong (expensive) one—Inter-Cascade enables the strong model to act as a “teacher” by generating reusable, generalized problem-solving strategies whenever it resolves a hard query. These strategies are stored in a lightweight, similarity-based memory and retrieved to augment the weak model’s input for future similar queries, allowing it to solve them locally via in-context learning. This eliminates redundant calls to the strong model for recurring or similar tasks. The approach requires no fine-tuning, works with both open-source and API-based LLMs, and probably improves calibration and system accuracy while significantly reducing cost and strong-model usage.

**Strengths:**

(1)This method enables continuous, online improvement of the weak model without fine-tuning—critical for API-based or resource-constrained settings.

(2)This paper provides a formal analysis showing that strategy-augmented inputs improve the weak model’s confidence calibration, leading to provably tighter accuracy guarantees under the same risk tolerance.


(3)This method is  compatible with any existing deferral function, model size, or cascade depth, and works seamlessly with both open-weight and closed API models.

(4)This study addresses a real-world inefficiency: Directly tackles the waste in standard cascades caused by repeated handling of similar hard queries—a common scenario in real user traffic (e.g., recurring math or factual questions).

**Weaknesses:**

（1）The proposed Inter-Cascade framework relies on prompt augmentation via a strategy repository, but it does not investigate whether more active learning mechanisms (e.g., periodic supervised fine-tuning) could yield stronger or more sustainable improvements.

（2）Experiments are confined to four English-language, closed-ended benchmarks (mostly mathematical reasoning and factual MCQs), limiting generalizability.

（3）The "strategy" is vaguely defined as a sequence containing the query, answer, and “generalized ideas,” with no ablation to determine which components drive performance gains.

（4）The paper claims low computational overhead for retrieval but omits end-to-end latency measurements and real-world deployment considerations (e.g., memory growth, cold-start effects).

（5）The theoretical guarantee (Theorem 2.2) assumes idealized improvements (error reduced by factor epsilon, coverage increased by factor b) without estimating these parameters from data.

**Questions:**

Could similar gains of inter cascade be achieved with static retrieval-augmented generation (RAG) using a pre-built knowledge base?

---

> ### Author Response · Authors · 2025-11-22
> **Response to Weaknesses:**
>
> We appreciate the reviewer’s insightful observations and constructive criticism.
>
> 1. **Comparison with Learning Mechanism**:
>
>     As we mentioned in **common response 6**, the motivations between Inter-cascade and Fine-tuning are different. Besides:
>
>     1) Fine-tuning is usually conducted in training phase while inter-Cascade work at inference time without training.
>
>     2) Fine-tuning is not cost friendly: Fine-tuning need prepare specialized dataset, extra long time for training, GPU power, (e.g., fine-tune Qwen3-235B requires 130 GB VRAM for LoRA or 2560 GB for full-parameter fine-tuning[1]
>
>     3) Fine-tuning for API-based LLMs is not available.
>
>     On the other hand, Inter-Cascade doesn’t conflict with fine-tuning. Inter-Cascade can be deployed on a fine-tuned model.
>
>     Reference: [1] Yang, An, et al. "Qwen3 technical report." arXiv preprint arXiv:2505.09388 (2025).
>
> 2. **Choice of Benchmark**:
>
>     The motivation of Inter-Cascade is to take advantage of the similarity that exists in some real-world applications. The reason why we tested the performance on those benchmarks is to verify that Inter-Cascade can fully exploit its advantages in scenarios with inherent similarity.
>
>     As mentioned in **common response 3**, we are currently running experiment on benchmarks that contains “diverse” samples suggested by reviewers. Our initial findings indicate that in our method, the overall accuracy is comparable to standard LLM Cascade but the weak LLM is still improved and the Call Strong Rate is still decreased, which imply we can always have better trade-off. We will update the evaluation once we finish the work on new benchmarks.
>
> 3. **Ablation on Strategy**:
>
>     we have added ablation study for the impact of each part of Weak LLM Input In Table 7 and Table 8 and added one paragraph to analyze the result. We test:
>
>     1) when the input for weak LLM is just [original Question, most similar question & strong answer] (No strategy),
>
>     2) when the input for weak LLM is [original Question, randomly selected question, strong answer, strategy] (Random)
>
>     3) our Inter-Cascade: [original Question, most similar question & strong answer, strategy].
>
>     The result shows that No strategy is harmful to weak LLM accuracy and will dramatically increase the usage frequency of strong model, leading to a relatively high overall accuracy, but contradicting with the demand of efficiency. Compared with Table 1, we can find that adding similar Q&A without generalized problem solving strategy will mislead weak LLM e.g weak LLM accuracy drops from 65.30% to 28.21. The method that chooses random strategy derives a result between our Inter-Cascade with standard LLM Cascade. The ablation experiments address the importance of adding the most similar strategy to the input of Weak LLM.
>
> 4. **Deployment Considerations**:
>
>     As mentioned in **common response 5**, we added thorough cost/latency/storage analysis results to Table 5 &6. In Table 6, we can find that compared to the time spent on answer generation, the retrieval  only occupies a tiny portion of total time (around 5%) and database size is small (10MB level) over 10k queries.
>
>     We are currently add experiments to measure the effect of cold-start, and will update once done.

---

> > ### Author Response · Authors · 2025-11-22
> > **(Continued) Response to Weakness**
> >
> > 5. **Clarification on Theorem**：
> >
> > Thank you for this insightful comment. We agree that Theorem 2.2 is derived under idealized assumptions, the improvement in error and coverage can be summarized by multiplicative factors $\epsilon$ and $b$. This abstraction
> > follows the formulation of the risk-tolerance function $\alpha$ introduced by [2], which limits our ability to characterize arbitrary forms of improvement analytically. Our theoretical result should therefore be viewed primarily as a qualitative and directional guarantee: it quantifies how risk tolerance changes if the error decreases by factor $\epsilon$ and the coverage increases by factor $b$, providing intuition about how improvements propagate through $\alpha$.
> >
> > To empirically ground these parameters, we estimated $\epsilon, b$, and the resulting theoretical change in $\alpha$ from our four benchmarks. We put them in the following table.
> >
> > | Quantity | GSM-Symb. | GSM-Plus | MetaMath | NASA-Hist. |
> > |---------|-----------|----------|----------|------------|
> > | ε       | 0.9179    | 0.9408   | 0.8854   | 1.0618     |
> > | b       | 1.7022    | 1.2579   | 1.2804   | 1.0565     |
> > | α_diff  | 3.39×10⁻³ | 1.67×10⁻³| 1.35×10⁻³| NaN        |
> >
> > Note: The NASA-Hist. benchmark does not satisfy the assumptions of Theorem 2.2 because $\epsilon>1$, and hence the change of $\alpha$ is not applicable.
> >
> > These empirical values show that the assumptions of Theorem 2.2 are not only interpretable but also estimable from real-world behavior of our system. Importantly, the predicted changes in $\alpha$ are small but consistently positive across three benchmarks that meet the assumptions of Theorem 2.2, matching the modest but quantifiable improvements the theory anticipates.
> >
> > More importantly, we want to emphasize that Theorem 2.2 provides a probabilistic upper bound on risk tolerance, not a prediction of realized accuracy. The practical gains achieved by our pipeline can significantly exceed the theoretical bound, as reflected in our tables. This is expected: the bound is conservative by design, whereas the cascade mechanism often amplifies improvements beyond the idealized multiplicative model.
> >
> > Reference: [2] Jung, Jaehun, Faeze Brahman, and Yejin Choi. "Trust or escalate: Llm judges with provable guarantees for human agreement." arXiv preprint arXiv:2407.18370 (2024).
> >
> > **Response to Questions:**
> >
> > 1. **Comparison with static RAG**:
> >
> >     For a pre-build knowledge base, it’s hard to define the scope of the database. It highly depends on the applications. A fixed database could handle specific classes of problems but
> >
> >     (1) need manually choose the database,
> >
> >     (2) not flexible to dynamic setting.  For example, if the weak LLM is deployed on the mobile device, one static database can’t handle the case when the input distribution is drifting like the user is moving to a different environment.
> >
> >     While in Inter-Cascade, the database is automatically generated through the interactions between weak LLM and strong LLM without human engagement and since we are keep updating the knowledge base according current type of questions, our method is  adaptive to the distribution of environment.

---

> > > ### Author Response · Authors · 2025-12-03
> > > **Additional Experiment Updated**
> > >
> > > We have added those experiments that you suggested, please also refer to the **point 8 and 9** in common response:
> > >
> > > 1. Continued Response to **2. Choice of Benchmark** (in first round): we added experiments on four more benchmarks. Table 10-13.
> > > 2. Continued Response to **4. Deployment Considerations** (in first round): We added a cold-start study and the result is shown in Figure 5. The result shows that at early stage, the pipeline accuracy for our Inter-Cascade is much close to baseline method. However, as the size of stored strategies increase, the performance of Inter-Cascade increase and gradually exceed baseline and eventually converges.
> > >
> > > **Lasted version of paper(v2) has been uploaded.**
> > >
> > > We deeply appreciate the time and effort you’ve dedicated to helping us refine and enhance our work!

---

### Official Review · Reviewer_9Qqx · 2025-11-02

**Soundness:** 2
**Presentation:** 2
**Contribution:** 2
**Rating:** 2
**Confidence:** 3

**Summary:**

The standard cascade defers difficult queries from a weak model to a strong model. This  work introduces an interactive loop: when the strong model handles a query, it also generates a "strategy" for solving it. This strategy is stored and retrieved for similar future queries to augment the weak model's input via in-context learning, thereby improving its  performance over time without explicit retraining. The authors present empirical results on four benchmarks, showing improvements in overall accuracy and a reduction in calls to the expensive strong model compared to a non-interactive cascade baseline.

**Strengths:**

1. The paper successfully engineers a system that effectively reduces reliance on expensive LLMs in a cascade setting. The empirical results, particularly the significant accuracy boost for the weak model on reasoning tasks and the reduction in API costs, demonstrate the practical utility of the proposed pipeline.
2. The framework's ability to allow the weak model to adapt and improve its performance online, without the need for costly fine-tuning, is a significant practical advantage. This  makes it applicable to scenarios involving proprietary, API-based models.
3. The analysis of how the method improves the weak model's confidence calibration is a valuable contribution. It demonstrates a deeper understanding of the system's dynamics beyond simple accuracy metrics, which is crucial for building reliable deferral systems.

**Weaknesses:**

1. The main weakness is that the paper fails to convincingly differentiate its contribution from a standard RAG system built on a dynamic knowledge base of past interactions. The core mechanism is retrieving textually similar past queries and their solutions to serve as few-shot prompts. This is not a fundamentally new idea.
2. The process for generating "strategies" is treated as a black box. The paper lacks crucial details on how the strong model is prompted to create solutions that are genuinely general and reusable, rather than just specific to a single query.
3. The cost analysis is superficial. It ignores the overheads introduced by the InterCascade framework itself: the token cost for generating the "strategy" (which is likely  longer than just an answer), the computational cost of the embedding and retrieval step for every query, and the storage costs of the growing repository.
4. The evaluation is heavily concentrated on structured mathematical reasoning problems. These tasks are particularly amenable to solution-pattern reuse. There is no evidence that the method is effective for more creative, nuanced, or structurally diverse tasks, which severely limits the claimed generality of the framework.

**Questions:**

1. What is the fundamental difference between your proposed "InterCascade" and a simpler baseline where a weak LLM uses a standard RAG module to retrieve from a dynamically updated vector database? Why was such a baseline not included in your evaluation?
2. Could you provide a formal cost for your framework that includes the token cost of strategy generation, the latency and computational cost of the retrieval, and the infrastructure cost of maintaining the repository? How do these overheads scale with query volume and repository size?
3. How do you expect your system to perform on tasks without clear, repeatable logical structures, such as creative tasks or multi-turn dialogue?

---

> ### Author Response · Authors · 2025-11-22
> **Responses to Weaknesses:**
>
> We appreciate the reviewer’s insightful observations and constructive criticism.
>
> Response to Weaknesses:
>
> 1. **The difference between Inter-Cascade and RAG**:
>
>     In short, the existing RAG approaches either
>
>     - (a) operate over a fixed, human-curated external corpus,
>     - (b) update a memory store using human dialogue and feedback, or
>     - (c) update a memory using self history for personalization without knowledge transfer.
>
>     To our knowledge, Inter-Cascade is the first framework where a weak LLM and a strong LLM jointly and autonomously build a RAG-like corpus. And the knowledge is transferred from a stronger LLM, not from the weak LLM itself.
>
>     We introduced the related types of RAG and made a comparison in **common response 4**. We also added a subsection for RAG, with detailed comparison with specific RAG-related papers, in Appendix B of the revised version 1 of paper.
>
>
> 2. **Strategy Template**: the complete prompt templates are attached in appendix I.
>
>     For the strategy part, we prompt that:
>     ``Start with \[Strategy\] section explaining the general approach for solving similar problems''. And then let LLM answer in this format:
>     ```
>     \[Question\]:{question}
>     [\Strategy\]:
>     \[Answer\]:
>
> 3. **Cost Analysis**:
>
>     We admit that adding a strategy to weak input would generate more tokens on Weak LLM. However, when the Weak LLM's accuracy is improved, the Strong Call Rate decreases, meaning fewer tokens will be needed for the Strong LLM. Since we think of the Weak LLM as owned by the user (e.g. running on their phone) and the Strong LLM is provided by a service which charges per token, this tradeoff is beneficial. Generating tokens on Strong LLM is more expensive than Weak LLM so in total, we can save 15.75%-52.15% on price across different benchmarks.
>
>     We added more detailed on the cost analysis including: total tokens, # of input tokens, and # of output tokens on both Weak LLM and Strong LLM, price charge, average time spent on retrieval and generation per query, and the size of database. All the result are attached in Table 5 & 6.
>
> 4. **Choice of Benchmarks**:
>
>     The motivation of Inter-Cascade is to take advantage of the similarity that exists in some real-world applications. The reason why we tested the performance on those benchmarks is to verify that Inter-Cascade can fully exploit its advantages in scenarios with inherent similarity.
>
>     As mentioned in **common response 3**, we are currently running experiment on benchmarks that contains “diverse” samples suggested by reviewers. Our initial findings indicate that in our method, the overall accuracy is comparable to standard LLM Cascade but the weak LLM is still improved and the Call Strong Rate is still decreased, which imply we can always have better trade-off. We will update the evaluation once we finish the work on new benchmarks.

---

> > ### Author Response · Authors · 2025-11-22
> > **Response to Questions**
> >
> > Since some questions are related to weakness, the weakness response for those parts also work here:
> >
> > 1. **Relations to RAG**:
> >
> >     1.1 *Difference*: Please refer to **common response 4**. And our **response to weakness 1**
> >
> >     1.2 *RAG baselines*:
> >
> >     -  If the database is static, then it’s hard to define the scope of the database. It highly depends on the applications. For example,For example, if the weak LLM is deployed on the mobile device, one static database can’t handle the case when the input distribution is drifting like the user is moving to different environment.
> >
> >     - If the database is dynamic, like we compare in extensive related work, current RAG works consider either updating the database according to human feedback, or store the single LLM’s self dialogue history for personalization. To our knowledge, no work considers using a stronger LLM’s strategy/feedback to automatically augment the weak LLM. We can add a method where weak LLM generates its own problem solving strategy for future use, but I think it’s trivial that our strong LLM’s strategy would definitely outperform the weak strategy.
> >
> > 2. Please refer to **response 3** in Weaknesses Response
> >
> >
> > 3. Please refer to **response 4** in Weaknesses Response

---

> > > ### Author Response · Authors · 2025-12-03
> > > **Additional Experiments Updated**
> > >
> > > We have added those experiments that you suggested, please also refer to the point 8 and 9 in common response:
> > >
> > > 1. Continued Response to **Choice of Benchmarks** (in first round): we added experiments on four more benchmarks. Table 10-13.
> > >
> > > **Lasted version of paper(v2) has been uploaded.**
> > >
> > > We deeply appreciate the time and effort you’ve dedicated to helping us refine and enhance our work!

---

### Official Review · Reviewer_PVAa · 2025-11-10

**Soundness:** 2
**Presentation:** 3
**Contribution:** 3
**Rating:** 4
**Confidence:** 4

**Summary:**

The authors propose a method to improve the performance of LLM cascade systems in terms of cost and accuracy. In a cascaded LLM system, a series of increasingly strong LLMs are used for a single query, with the stronger LLMs employed only if the weaker LLM does not produce a confident enough answer. The work proposes storing the reasoning/strategy of the strong LLM into a repository. For a new query, the ‘k’ most relevant strategies from this repository are used as context to help the weak LLM. The authors experiment on reasoning and knowledge benchmarks and empirically show that the proposed method results in fewer queries being processed by the stronger LLM, while increasing the accuracy of the overall pipeline. This results in a decrease in the overall cost.

**Strengths:**

The proposed method is simple and intuitive. The solution of using stronger LLM’s reasoning as context for weaker LLM is interesting. The authors theoretically show that, for the proposed method, the risk tolerance $\alpha$ is reduced under certain assumptions. Empirical results consistently show that the proposed method results in higher overall accuracy and lower overall costs, along with increased accuracy for weaker LLM and fewer calls to stronger LLM.

**Weaknesses:**

The work lacks experiments. All experiments are performed with a single combination of just two LLMs (weak=Gemini-2.5, strong=GPT-3.5 turbo). The weak LLM’s performance is quite low and the gap between the weak and strong LLMs is high on all the math reasoning datasets. Would we still see significant improvements in accuracy and cost if the weak LLM performance is higher and the gap between the weaker and stronger LLMs is small?

Experiments are also limited to three math/logical reasoning benchmarks and a knowledge benchmark. The GSM-Symbolic and GSM-Plus datasets contain repeated instances of math questions with nearly the exact same wording but with the variable values modified. The proposed method is expected to perform the best in such setups but this does not represent real-world applications. How well does the strategy retriever work when the queries are significantly different from each other? Would you need more than just top-2 strategies as context in such cases? How would that impact the performance of the model and the overall cost? How well does the approach work for datasets where the queries are quite diverse not just in terms of wording but in terms of strategy required to solve them and there are no repeated queries with the exact same strategy? How well does the method work on reasoning benchmarks apart from math/logical reasoning? Does the performance translate to multi-modal datasets? The current datasets are quite small (a few thousand samples). Would the retrieval quality and thus the overall performance suffer when scaled to much larger datasets?

The work requires more experiments with different benchmarks and LLMs are necessary to answer these questions.

**Questions:**

1. Provide results on additional benchmarks (e.g., the typically used 8 benchmark datasets for common-sense reasoning [a]) and more challenging benchmarks like Big-Bench Hard [b], MMLU-Pro [c] and AGIEval [d].
2. Provide results with different weak-strong LLM combinations including those with a good weak LLM performance and lower performance gap between weak and strong LLMs.
3. How does the number of strategies retrieved (‘k’) affect performance and cost, particularly on benchmarks without repeated/highly similar queries?
4. Provide more discussion / specific examples of real-world applications where the proposed method would be useful (setups where the queries and solution strategies are highly similar to one another).
5. The proposed method likely reduces the time required per query significantly. Provide results on throughput with and without the proposed approach.
6. How well does the method scale to really large and diverse datasets in terms of both efficiency and task performance?


References:

[a] Hu, Zhiqiang, et al. "Llm-adapters: An adapter family for parameter-efficient fine-tuning of large language models." Proceedings of the 2023 conference on empirical methods in natural language processing. 2023. \
[b] Suzgun, Mirac, et al. “Challenging BIG-Bench Tasks and Whether Chain-of-Thought Can Solve Them.” arXiv preprint arXiv:2210.09261, 2022. \
[c] Wang, Yubo, et al. “MMLU-Pro: A More Robust and Challenging Multi-Task Language Understanding Benchmark.” arXiv preprint arXiv:2406.01574, 2024.\
[d] Zhong, Wanjun, et al. “AGIEval: A Human-Centric Benchmark for Evaluating Foundation Models.” Findings of the Association for Computational Linguistics: NAACL 2024, 2024, pp. 2299-2314.

---

> ### Author Response · Authors · 2025-11-22
> **Responses to Weakness:**
>
> We appreciate the reviewer’s insightful observations and constructive criticism.
>
> **Responses to Weakness**：
>
> 1. **Choice of LLMs**: One of the motivation for LLM Cascade diagram is that strong model is accurate but costly while weak model is relative inaccurate but cheap. Jointly using them can reach a better trade-off between cost and accuracy. The existing gap between the weak LLM and strong LLM validate the advantage of LLM Cascade.  But of course, we are willing to test on different pairs of weak LLM and Strong LLM to provide a more comprehensive analysis for our method Inter-Cascade, we are currently conducting this experiment and will update once finished.
>
> 2. **Choice of Current Benchmarks**:
>
>     2.1 As mentioned in common response 3 **Choice of benchmarks**, our choice of benchmarks which contains variants are not trivial. these are harder versions of the original benchmarks. For example: significant performance drops (up to 65%) are observed on GSM-symbolic across all state-of-the-art models compared to GSM8K and performance drops (up to 40.56%) are observed in GSM-Plus compared to GSM8K . Our results show that Inter-Cascade offers significant improvements compared to standard LLM Cascade. Improvement on those benchmarks is still meaningful since there are a lot applications that concentrate on small set of tasks as mentioned in common response 1 **Motivation of our work**.
>
>     2.2 *strategy retriever*: We are running experiments on benchmarks that are more "diverse". We will update the results once finished.
>
>
> 3. **Choice of top-2 strategies**: We think in any system that uses retrieval, it’s an open problem which highly depends on the application setting and specification of LLM. For example, if the context window for weak LLM is small, choosing top-k, when k is too large, would exceed the context limit. Or if the question is  more about factual check like a detail in a novel, then the context length for each strategy will be longer.   Therefore, we think the choice of k is more empirical. In our early stage testing, the weak LLM accuracy first increase and then decrease with the growth of k. Based the results on our benchmarks, we choose k =2.
>
>
> 4. **Diverse Benchmarks**: as mentioned in common response 3. We are running additional experiments on additional benchmarks that were suggested by reviewers and will update our response once those evaluations are complete. Our initial findings indicate that in our method, the overall accuracy is comparable to standard LLM Cascade on benchmarks that are more “diverse”, but with the help of retrieved strategy, the weak LLM’s accuracy and confidence are still improved. Therefore, the Call Rate of strong LLM is still much less than the standard LLM Cascade, which demonstrates that Inter-Cascade provides a better trade-off between accuracy and cost on various benchmarks.
>
> 5. **Multi-modal**: our diagram is general, for example, the similarity function is not limited text, can be extended to figure. in VLM, image is also transformed as embedding, which is similar to text. We envision the application of autonomous driving: on-vehicle VLM would send image to strong VLM on cloud for rare or long-tail event, the strong VLM identify the image and send back action decision and together with an event summary (strategy) and the on-vehicle VLM can get improved in future driving.
>
> 6. **Performance on Larger dataset**:
>
>     We scale up the testing size of MetaMath from 10K to 20K:
>
>     - Pipeline accuracy: 70.76% -> 71.56%
>
>     - Strong Call Rate: 34.98% -> 23.68%
>
>     - Weak Accuracy: 50.31% -> 58.16%
>
>     The pipeline accuracy is close to each other  (+0.8%), the strong Call Rate decreases a lot (-11.3%)and the accuracy on Weak LLM increase (7.85%). The results make sense because along with the increase of query and defer, more strategies are generated by a strong model. As more topics are covered, the retrieved strategies will be more relevant to the question and more helpful, which reflect at the increase of Weak LLM accuracy. Since the weak LLM is more reliable with the help of strategies, the need to call strong LLM is less, so the Call Strong Rate decreases. The retrieval framework we used is FAISS[1], which is a well developed open-source that has been used on a lot large-scale recommendation system and their user volume scale to millions. So we think the size of datasets won’t the bottleneck of our methods.
>
> [1]Douze, Matthijs, et al. "The faiss library." IEEE Transactions on Big Data (2025).

---

> ### Author Response · Authors · 2025-11-22
> **Responses to Questions:**
>
> Since some questions are related to weakness, the weakness response for those parts also work here:
>
>
> 1. Please refer to **response 4** in Weaknesses Response
>
> 2. Please refer to **response 1** in Weaknesses Response
>
> 3. We are running experiments with different number of strategy, will update once finished
>
> 4. Please refer to **common response 1 and 2**
>
> 5. We added more detailed on the cost analysis including total, input, output tokens on both Weak LLM and Strong LLM, price charge, average time spent on retrieval and generation per query, size of database. The current cost analysis tables are in Table 5 and Table 6 of our revised version paper.
>
> 6. Please refer to **response 6** in Weaknesses Response.
>
> **Paper Updating**:
> Revised Version 1 of Paper has been Updated to OpenReview.

---

> > ### Author Response · Authors · 2025-12-03
> > **Additional Experiments Updated**
> >
> > We have added those experiments that you suggested, please also refer to the **point 8 and 9** in common response:
> >
> > 1. Continued Response to **1. Choice of LLMs** (in first round): We have tested on different Weak LLM (set to Gemini 2.0) and Strong LLM (set to Gemini 2.5) pairs. The results are attached in Appendix K3 and Table 14, 15, and 16.
> >
> > 2. Continued Response to **3. Choice of top-2 strategies** (in first round): We adde an ablation study to show the effect of strategy number on the performance in Figure 6.
> >
> > 3. Continued Response to **4. Diverse Benchmarks** (in first round): we added experiments on four more benchmarks. Table 10-13.
> >
> > **Lasted version of paper(v2) has been uploaded.**
> >
> > We deeply appreciate the time and effort you’ve dedicated to helping us refine and enhance our work!

---

### Author Response · Authors · 2025-11-22
**Response: common points brought up by reviewers**

We thank the reviewers for their extensive comments regarding our work. We wish to first clarify a few points which were common to many reviews. We integrated these responses into a revised version of the paper.

1. **Motivation of our work**:

    Efficiency is a key challenge for the wider deployment of LLMs and cascade architectures offer a way to use the full power of very large models selectively as needed. For example, last week’s release of GPT 5.1 improves on GPT 5.0 by allowing the model to adaptively spend more time on complex problems (GPT-5.1 Thinking mode) and less on simpler problems (GPT 5.1 Instant mode) [1]. Many envisioned future applications will use a weaker Small Language Model (SLM) that can defer to a stronger LLM on difficult instances. NVIDIA’s recent position paper [2] claims that SLMs will be a key component of Agentic AI:  many applications repeatedly perform a small set of specialized tasks with modest variations between tasks.

    We take this view as a starting point and propose adding a feedback mechanism to the standard LLM Cascade approach of deferring from weak LLM to strong LLM. In our paradigm the strong LLM can act as a teacher for the weak LLM to make it more efficient on repetitive tasks. This approach can have significant benefits for latency, memory, computation, and cost, particularly when the weak LLM is free while the strong LLM charges (typically per token).

2. **Application Domains of Interest** :

    The primary class of applications which we are targeting are those in which one model is smaller or “weaker” due to local constraints. A prototypical example is mobile devices running on a battery: the resources needed (time, memory, energy) by a strong LLM may be prohibitive. Future wireless systems envision the small model deferring to a remote server (sometimes called the mobile edge cloud or MEC) that has the LLM. Deferral can create higher latency and expense, particularly if the strong LLM is operated by a third party.

    Another example of repetitive workloads come from help or customer service system agents. While the bulk of questions may be simple, these systems may occasionally be presented with complex and unexpected queries. In this case deferring using our approach can improve the performance of the help system in the short to medium term. The full agent model can be updated/replaced on a slower time scale (a periodic software update).

    Inter-Cascade is best suited for applications where a SLM is good enough for a large portion of queries but there is still a significant set of queries for which it was not pretrained and would need help. In a sense we are adapting to the local input distribution.

    The metaphor we are using is akin to that of a student (querier) interacting with a TA (an SLM) who can defer to the professor (an strong LLM). If a student needs help on a problem and the TA does not know, the instructor would give not only the answer but also the steps to solve the problem (the strategy).

3. **Choice of benchmarks**:

    The benchmarks we used in our submitted manuscript exhibit the features of input instances for which Inter-Cascade was designed: workloads which contain common questions with variations. For the reasoning-focused scientific benchmarks, we choose GSM-symbolic, GSM-Plus and Metamath. They are variants from GSM8K and MATH but the variants are non-trivial: these are harder versions of the original benchmarks. For example: significant performance drops (up to 65%) are observed on GSM-symbolic across all state-of-the-art models compared to GSM8K shown in Fig. 3 in [3] and performance drops (up to 40.56%) are observed in GSM-Plus compared to GSM8K shown in Table 4 in [4]. Our results show that Inter-Cascade offers significant improvements compared to standard LLM Cascade.

    As pointed out by the reviewers, these are relatively specialized reasoning problems with fairly “obvious” similarities between queries. We are therefore running additional experiments on additional benchmarks that were suggested by reviewers and will update our response once those evaluations are complete.  Our initial findings indicate that in our method, the Overall Accuracy achieves a comparable performance compared with LLM Cascade on benchmarks that are more “diverse”, while still achieving superior performance on benchmarks that contain more similarity. But for both types, the weak LLM’s accuracy and confidence are always improved. Therefore, the Call Rate of strong LLM is always less than the standard LLM Cascade, which demonstrates that Inter-Cascade provides a better trade-off between accuracy and cost on various benchmarks. Comprehensive results on new benchmarks will be updated once the experiments are done.

---

> ### Author Response · Authors · 2025-11-22
> **Response (continued): common points brought up by reviewers**
>
> 4. **Relationship to RAG**:
>
>     The reviewers all had questions about how our solution relates to retrieval augmented generation (RAG). It is true that our solution concept is similar in flavor to RAG. According to the relationship to our diagram, the approaches typically described as RAG largely fall into three categories:
>
>     **Static RAG** is given a fixed (pre-set) database, how can we efficiently retrieve, rank, and merge information from the database to create a more effective query? This is the original RAG problem. [10][11]
>
>     Inter-Cascade builds the database on the fly rather than starting with an existing and fixed database.
>
>     **History-aware RAG** is a way of using human feedback (e.g. in an interactive chat) with a single LLM to augmen the database for future interactions (see e.g. [6][7]). The decision of when to update is either triggered by the user [6] and the user decides what should be added, or is automatically decided by the LLM [7] as a form of self-improvement. The long-term effect is that the LLM is customized or personalized to the user.
>
>     In Inter-Cascade, the decision to update is made by the weak LLM and the content of the update is decided by the strong LLM: the user is not part of the loop. This is appropriate when a single system interacts with many users (for example as a help desk).
>
>     **Agentic RAG** makes the database a resource for one or more agents [8][9]. A single agent is more like Static RAG. In multiple agent cases, the different agents are assigned with different roles in the system, like one for planner, one for retriever with the database and one for answer/action generations. The interaction between agents doesn’t contribute to the update of the RAG database.
>
>     In short, most RAG approaches use either (i) operate over a fixed, human-curated external corpus, (ii) update a memory store using human dialogue and feedback, or (iii) update a memory using self history for personalization without knowledge transfer. In Inter-Cascade, the database (as such) is built on the fly through interactions between the weak LLM and strong LLM. To our knowledge, ours is the first work to actively use the strong LLM to build a more effective corpus/database in a Cascade system.
>
>     While we do not believe our approach is simply a form of RAG, we will be more explicit about these connections so the reader will be clearer on the similarities and differences. A more detailed comparison between RAG and our method is attached in our revised version paper in Appendix B: Extended Related Work Section.
>
> 5. **More fine-grained cost analysis**:
>
>     We agree with the reviewer that providing a more detailed breakdown of the costs can help explain the benefits and tradeoffs of Inter-Cascade. After running additional experiments on current benchmarks we have provided updated tables with these breakdowns to allow for a more nuanced understanding. The current cost analysis tables are in Table 5 and Table 6 of our revised version paper.
>
> 6. **Relationship to fine-tuning**:
>
>     It is natural to think of Inter-Cascade and fine-tuning as trying to accomplish similar goals. However, the motivations are different: in fine-tuning the goal is to take a performant model and specialize it to a particular domain, typically using offline training. The domain is typically known or a training dataset must be generated for that domain. Inter-Cascade works by improving a less performant model to specialize it for the workload that appears at inference time: there is no training. This has the advantage of not requiring access to model internals (such as embeddings) and is more compatible with API access to the strong model. On the other hand, Inter-Cascade doesn’t conflict with fine-tuning. Inter-Cascade can be deployed on a fine-tuned model.

---

> > ### Author Response · Authors · 2025-11-22
> > **Response (continued): common points brought up by reviewers**
> >
> > 7. **List of Modification on  revised version 1 of paper**:
> >
> >     + Refined abstract and introduction, add more related works and comparison.
> >
> >     + Ablation on no-strategy: reviewer jo4V suggested we conduct ablation studies to evaluate the impact of strategy. We integrate the method with random strategy from Table 3&4, together with method which only patch similar Q&A (without strategy) to input. And put the results in new Table 7&8.
> >
> >     + Cost analysis: add more details in Table 5 and create Table 6 for latency result
> >
> >     + Since reviewer suggested to add test on larger datasets, we updated our result on MetaMath with 20k samples (was 10K).
> >
> >     + Added one more subsection on comparison with RAG in Appendix B: extended related work
> >
> >
> > Reference:
> >
> > [1] https://openai.com/index/gpt-5-1/
> >
> > [2] Belcak, Peter, et al. "Small Language Models are the Future of Agentic AI." arXiv preprint arXiv:2506.02153 (2025).
> >
> > [3]Mirzadeh, Iman, et al. "Gsm-symbolic: Understanding the limitations of mathematical reasoning in large language models." arXiv preprint arXiv:2410.05229 (2024).
> >
> > [4]Li, Qintong, et al. "Gsm-plus: A comprehensive benchmark for evaluating the robustness of 。llms as mathematical problem solvers." arXiv preprint arXiv:2402.19255 (2024).
> >
> > [5]Yang, An, et al. "Qwen3 technical report." arXiv preprint arXiv:2505.09388 (2025).
> >
> > [6]Chen, Qinwen, et al. "ComRAG: Retrieval-Augmented Generation with Dynamic Vector Stores for Real-time Community Question Answering in Industry." arXiv preprint arXiv:2506.21098 (2025).
> >
> > [7]Zhang, Feiyuan, et al. "Dh-rag: A dynamic historical context-powered retrieval-augmented generation method for multi-turn dialogue." arXiv preprint arXiv:2502.13847 (2025).
> >
> > [8]Nguyen, Thang, Peter Chin, and Yu-Wing Tai. "MA-RAG: Multi-Agent Retrieval-Augmented Generation via Collaborative Chain-of-Thought Reasoning." arXiv preprint arXiv:2505.20096 (2025).
> >
> > [9]Li, Yangning, et al. "Towards agentic rag with deep reasoning: A survey of rag-reasoning systems in llms." arXiv preprint arXiv:2507.09477 (2025).
> >
> > [10]Lewis, Patrick, et al. "Retrieval-augmented generation for knowledge-intensive nlp tasks." Advances in neural information processing systems 33 (2020): 9459-9474.
> >
> > [11]Edge, Darren, et al. "From local to global: A graph rag approach to query-focused summarization." arXiv preprint arXiv:2404.16130 (2024).

---

> ### Author Response · Authors · 2025-12-03
> **Second Round Response: Additional Experiments Updated**
>
> 8. **Complementary response for 3. Choice of benchmarks (1st round response):**
>
>
> We have tested and added the results on four more benchmarks (now eight benchmarks in total):
>
> + GSM8K [1] (well-known datasets suggested by reviewer qpUq)
> + Big-Bench Hard [2] (more challenging benchmark suggested by reviewer PVAa)
> + BarExam [3] (law question answering benchmark. Evaluated according to the task diversity suggestion by reviewer 9Qqx and jo4V)
> + MedMC [4] (medical question that covers 2400 healthcare topics and 21 medical subjects. Evaluated according to the task diversity suggestion by reviewer 9Qqx and jo4V)
>
> The results of pipeline accuracy and Strong LLM Call rate are shown in Table 10. The results of weak LLM performance are shown in Table 11. The cost analysis and latency analysis are shown in Table 12 and Table 13 as well. Since we extended the size of testing benchmarks from four to eight, we don’t have enough space to put all the results in the main part of the paper, the results on new benchmarks are put in Appendix J.
>
> For those four benchmarks: our method outperforms the baseline method, with a 0.18% - 3.96% increase in Pipeline Accuracy. The Strong LLM Call Rate is reduced on all benchmarks, with reductions ranging from 1.52% to 16.14%. The improvements on the accuracy for the samples that are predicted by Weak LLM are between 2.24% and 15.56%. The money spent on API queries are saved by 2.33%-83.94%.  Although Inter-Cascade is not targeted to improve system performance for the scenarios that do not contain explicit similarity or variants (as a comparison, see note below), our method can still achieve at least comparable pipeline accuracy for general benchmarks. Meanwhile, our method still improves the accuracy and confidence of Weak LLM, and reduces the usage of Strong LLM, consequently, saving money. **Therefore, in those general tasks, Inter-Cascade still provides a better trade-off between accuracy and cost compared to the baseline method.** One explanation is that similarity is a nature that exists in any task or benchmark. Our Inter-Cascade mechanism helps elicit the latent similarity and improve the performance of Weak LLM through the interaction between Weak LLM and Strong LLM.
>
>
> > **Note:** Inter-Cascade is more beneficial for benchmarks that contain explicit variants: In GSM-Symbolic, GSm-Plus and MetaMath, our method outperforms the baseline method, with a 4.33% - 6.35% increase in Pipeline Accuracy. The Strong LLM Call Rate is reduced 13.85% to 28.53%. The improvements on the accuracy for the samples that are predicted by Weak LLM are between $6.2% and 33.06%.
>
>
>
>
> 9. **Extra Ablation Study** To better evaluate the performance and generalization capacity of Inter-Cascade, we set up extra ablation studies:
>
> + The dynamic of pipeline accuracy along with the number of queries compared with baseline. The results are attached in Appendix K1 and Figure 5. The result shows that at early stage, the pipeline accuracy for our Inter-Cascade is close to baseline method. However, as the size of stored strategies increase, the performance of Inter-Cascade increase and gradually exceed baseline method and eventually converges.
> + The effect of strategy number on the pipeline accuracy. The results are attached in Appendix K2 and Figure 6. The results show that the trend of pipeline accuracy is increasing first, reaching peak and then decreasing along with the number of strategies. The result makes sense because too few strategies might not retrieve the best strategy in repository, while too many strategies might distract the answer from certain query question. In our experiment, the empirical best number of strategies $k$ is 2.
> + The result of different Weak LLM (set to Gemini 2.0) and Strong LLM (set to Gemini 2.5) pairs. The results are attached in Appendix K3 and Table 14, 15, and 16. The trend and conclusion still coincide with the original pair of Weak LLM and Strong LLM.
>
> Reference:
>
> [1] Cobbe, Karl, et al. "Training verifiers to solve math word problems." arXiv preprint arXiv:2110.14168 (2021).
>
> [2] Suzgun, Mirac, et al. "Challenging big-bench tasks and whether chain-of-thought can solve them." Findings of the Association for Computational Linguistics: ACL 2023. 2023.
>
> [3] Zheng, Lucia, et al. "A reasoning-focused legal retrieval benchmark." Proceedings of the 2025 Symposium on Computer Science and Law. 2025.
>
> [4] Pal, Ankit, Logesh Kumar Umapathi, and Malaikannan Sankarasubbu. "Medmcqa: A large-scale multi-subject multi-choice dataset for medical domain question answering." Conference on health, inference, and learning. PMLR, 2022.

---

> ### Author Response · Authors · 2025-12-03
> **Final Version (v2) of Paper in Rebuttal Session Uploaded**
>
> Dear ACs and Reviewers,
>
> Thank you very much for your valuable comments and suggestions. We have carefully addressed each point you mentioned and incorporated all revisions into the **latest version of the paper (v2)**. We understand this is a particularly challenging time, and although you may not be able to respond because of the new rule, we hope that our efforts and response have resolved your concerns.
>
> We wish you all the best, and thank you again!
>
>
>
> Checklist for the revision:
>
> + Original version: 28 pages. Latest version 37 pages.
> + Refined Abstract and Introduction, add more related works and comparison.
> + Conducted experiments on 8 various benchmarks (added 4 during rebuttal): 1) GSM8K, 2)Big-Bench Hard, 3)BarExam 4)MedMC, 5)NASA-history, 6) GSM-Symbolic, 7) GSM-Plus, 8) MetaMATH. New Results are shown in Table 9-13, Appendix J. **The results show that Inter-Cascade provides better trade-off on accuracy and cost compared to standard LLM Cascade across all benchmarks.**
> + Ablation Study:
>     + Ablation on no-strategy: one reviewer suggested we conduct ablation studies to evaluate the impact of strategy. We integrate the method with random strategy from Table 3&4, together with method which only combine similar Q&A (without strategy) to input. And put the results in new Table 7&8. **The results show that adding strategy to Weak LLM input is a significant factor on the improvement.**
>     + Cold Start of Strategy repository: Evaluated the dynamic of pipeline accuracy along with the increasing number of queried samples. Result is shown in Figure 5. The performance of Inter-Cascade would converge regardless of the increasing size of strategy repository.
>     + Effect of the Strategies Number: Evaluated the impact of number of retrieved strategies on the pipeline accuracy. Result is shown in Figure 6. The results are not monotone to the number of retrieved strategies. The best choice of k can be measured empirically.
>     + New Weak LLM and Strong LLM pairs: to show the Inter-Cascade is general to any pair of Weak LLM and Strong LLM, we tested on Gemini2.0-flash and Gemini2.5-flash pair. The Results are shown in Table 14-16. **The conclusions for the original pair also work for the new pair. Inter-Cascade is a general framework.**
> + Cost analysis: add more details in Table 5 and create Table 6, 12 and 13 for cost and latency result. **Considering the improvement on accuracy and save on money, the tiny memory cost and extra retrieval time are acceptable and even negligible.**
> + Since reviewer suggested to add test on larger datasets, we updated our result on MetaMath with 20k samples (was 10K).
> + Added one more subsection on comparison with RAG in Appendix B: extended related work. **Inter-Cascade is different from existing RAG pipeline**.

---

### Comment · Area_Chair_fgtk · 2025-11-24

Dear Reviewers and Authors,

**Authors** Thank you for your response. **Reviewers** Please read the authors rebuttals and the other reviews to start the discussion. Please respond as early as possible to give everyone enough time to engage in the discussion.

Best,

AC

---

### Meta-Review · Area_Chair_zETn · 2026-01-13

**Summary:**

This paper considers solving problems using a cascade of LLMs. Instead of simply deferring challenging questions to stronger LMs, this work proposes that the strong LMs store the example with reasoning strategy to a database that can later be queried to be used as context to help weak LMs. Experiments show higher accuracy and lower costs compared to baselines.

Overall, reviewers score this paper with 4/2/4/6, and the main concerns are the limited experiment settings such as only one strong LM and one weak LM are used, the lack of datasets beyond closed-ended math tasks, the scalability of the approach when the database grows in size, and also the differentiation with existing RAG systems. Given the overall reviewer opinions, I'm not recommending the acceptance of the paper in its current form.

**Reviewer Concerns:**

1.the concern about the lack of more stronger/weaker LMs is partly addressed with additional experiments.
2. the concern about more diverse datasets / tasks is partly addressed as the authors scaled up the evaluation on metamath.
3. the concerns about baselines, presentation clarity, and related work are mostly addressed.
4. the concern about extra costs due to the retrieval step when the database grows large is partly addressed (but not clear whether this scales to larger scales).

**Reviewer Scores:**

1. PVAa: due to the concern about more diverse datasets is not fully addressed, the reviewer is likely to keep their score.
2. 9Qqx (marked as AI-generated): the concern about the difference with respect to RAG is addressed, the concern about the lack of more diverse tasks seems is not fully addressed. It is possible that this reviewer might increase their current score of 2 to 3 or 4.
3. jo4V: the concerns mostly overlap with 9Qqx, and I think it is likely that the reviewer will keep their score.
4. qpUq: concerns about baselines, presentation clarity, and related work are mostly addressed.

---

### Decision · Program_Chairs · 2026-01-26

Reject